# An HDAC6-dependent surveillance mechanism suppresses tau-mediated neurodegeneration and cognitive decline

Hanna Trzeciakiewicz [1], Deepa Ajit[2], Jui-Heng Tseng [2], Youjun Chen[2], Aditi Ajit[2], Zarin Tabassum [2], Rebecca Lobrovich[3], Claire Peterson[3], Natallia V. Riddick [4], Michelle S. Itano [5], Ashutosh Tripathy[1], Sheryl S. Moy[6], Virginia M. Y. Lee[7], John Q. Trojanowski [7], David J. Irwin[3] & Todd J. Cohen [1,2✉]

Tauopathies including Alzheimer's disease (AD) are marked by the accumulation of aberrantly modified tau proteins. Acetylated tau, in particular, has recently been implicated in neurodegeneration and cognitive decline. HDAC6 reversibly regulates tau acetylation, but its role in tauopathy progression remains unclear. Here, we identified an HDAC6-chaperone complex that targets aberrantly modified tau. HDAC6 not only deacetylates tau but also suppresses tau hyperphosphorylation within the microtubule-binding region. In neurons and human AD brain, HDAC6 becomes co-aggregated within focal tau swellings and human AD neuritic plaques. Using mass spectrometry, we identify a novel HDAC6-regulated tau acetylation site as a disease specific marker for 3R/4R and 3R tauopathies, supporting uniquely modified tau species in different neurodegenerative disorders. Tau transgenic mice lacking HDAC6 show reduced survival characterized by accelerated tau pathology and cognitive decline. We propose that a HDAC6-dependent surveillance mechanism suppresses toxic tau accumulation, which may protect against the progression of AD and related tauopathies.

[1] Department of Biochemistry and Biophysics, University of North Carolina, Chapel Hill, NC 27599-7260, USA. [2] Department of Neurology, UNC Neuroscience Center, University of North Carolina, Chapel Hill, NC 27599-7250, USA. [3] Penn Digital Neuropathology Laboratory, Department of Neurology, Perelman School of Medicine, University of Pennsylvania, Philadelphia, PA 19104-4283, USA. [4] Division of Comparative Medicine, University of North Carolina, Chapel Hill, NC 27599-7146, USA. [5] Department of Cell Biology and Physiology, Carolina Institute for Developmental Disabilities, UNC Neuroscience Center, University of North Carolina, Chapel Hill, NC 27599-7250, USA. [6] Department of Psychiatry, Carolina Institute for Developmental Disabilities, University of North Carolina, Chapel Hill, NC 27599-7146, USA. [7] Department of Pathology & Laboratory Medicine, Center for Neurodegenerative Disease Research (CNDR), Institute on Aging, Perelman School of Medicine, University of Pennsylvania, Philadelphia, PA 19104-4283, USA. ✉email: toddcohen@neurology.unc.edu

Tau inclusions form the hallmark pathological lesions observed in Alzheimer's disease (AD) and related tauopathies. The traditional view of tau pathogenesis has focused predominantly on hyperphosphorylated tau, but mounting evidence now implicates aberrant tau acetylation on lysine residues spanning the microtubule (MT) binding repeat region (MTBR) as another contributing factor[1–4]. The sheer abundance of tau acetylation within the MTBR (>25 lysine residues identified) and the ability of this modification to control tau binding to MTs, tau aggregation, and tau oligomer formation suggest that aberrant tau acetylation acts in a pathological manner to promote neurotoxicity and cognitive decline. Supporting this notion, recent studies indicate that acetylated tau is also sufficient to promote synaptic degeneration and cognitive dysfunction[3,5,6].

Analysis of diseased human postmortem brain tissues showed robust accumulation of acetylated tau within neurofibrillary tangles (NFTs) and other aggregated tau lesions. For example, using site-specific tau acetylation antibodies, we and others have found acetylated tau in AD brains and a range of other tauopathies including Pick's disease (PiD), corticobasal degeneration (CBD), and progressive supranuclear palsy (PSP)[1,7,8]. In contrast to phosphorylated tau, acetylated tau at residue K280 (ac-K280) is undetectable in cognitively normal control brains[1], suggesting ac-K280 is a disease-specific marker for tauopathies. This concept has important clinical implications since acetylated tau could be directly linked to tauopathy onset, tau strain specificity, and/or disease progression. Whether or not different tauopathies are characterized by distinct tau post-translational modification (PTM) profiles remains unclear. Such a scenario is certainly plausible given the unique structural, conformational, and seeding properties of tau in AD compared to other tauopathies[9–12].

The histone deacetylase (HDAC) family has been implicated in tau deacetylation[1,13]. In particular, the deacetylase HDAC6 is predominantly cytoplasmic, associates with MTs, and reversibly regulates tubulin acetylation, as well as other cytoplasmic substrates[14]. Selective HDAC6 inhibition has shown varying degrees of neuroprotection in some tauopathy or amyloid-beta (Aβ) plaque models[15–19]. For example, acute pharmacological HDAC6 inhibition with tubastatin A (TBST) alleviated behavioral and cognitive deficits in transgenic tau or Aβ mouse models[18,19], spurring significant interest in using HDAC6 inhibitors as AD therapies. However, we previously showed that acetylated tau is a substrate for HDAC6 and that inhibition of HDAC6 could result in elevated levels of acetylated tau[1]. Thus, chronic long-term HDAC6 inhibition or depletion could conceivably increase acetylated tau, promote tau aggregation, and lead to cognitive defects. Consistent with this possibility, some studies have reported a neuroprotective role for HDAC6 in protein quality control (PQC)[20–24], including the ability to target misfolded proteins and damaged organelles for degradation[25,26]. Interestingly, there are no cognitive abnormalities in HDAC6 KO mice[27], suggesting HDAC6 function is dispensable under normal physiological conditions. However, genetic manipulation of HDAC6 in the context of neurodegenerative tauopathies has not been examined until this study.

Broad spectrum HDAC inhibitors, including the widely used pan-HDAC inhibitor trichostatin A (TSA), differentially target class I (HDAC1/2/3/8), class IIa (HDAC4/5/7/9), class IIb (HDAC6/10), class III (SIRT1-7), or class IV (HDAC11) HDACs. Among the pan-HDAC inhibitors, several are FDA-approved and in various stages of clinical development for the treatment of neurological conditions as well as cancers (e.g., T-cell lymphoma and multiple myeloma)[28,29]. In addition to neurological applications, HDAC inhibitors are currently being developed for inflammatory diseases, muscular dystrophies, HIV/AIDS,

and more recently as protective agents against myocardial infarction[30,31]. Here, we directly addressed the significance of HDAC6 in the context of neurodegenerative tauopathies. We uncovered a novel mechanism through which HDAC6 suppresses aberrant tau accumulation, highlighting the loss of HDAC6 as a potential trigger for tau-mediated neurodegeneration.

## Results

**An HDAC6-chaperone complex targets tau via the MTBR domain.** We analyzed the interaction between full-length wild-type (WT) human tau and a panel of FLAG-tagged (FL) deacetylases, including class I, II, and III HDACs that are reported to localize and/or shuttle into the cytoplasm (Supplementary Fig. 1a, b). Co-immunoprecipitation (co-IP) assays revealed robust tau binding to HDAC3, HDAC6, and SIRT1 (Fig. 1a), but not other deacetylases including HDAC1, a predominantly nuclear class I HDAC (Supplementary Fig. 1c). Since HDAC6 bound to lower molecular weight dephosphorylated tau bands compared to the other deacetylases, this suggested that HDAC6 may be involved in the suppression of tau hyperphosphorylation and prompted us to further evaluate this particular interaction.

To assess the critical domains of HDAC6 that mediate tau binding, co-IP assays were performed using a panel of HDAC6 mutants lacking catalytic activity at one (H216A or H611A) or both deacetylase domains (H216/611A), lacking the entire C-terminus (1–840), lacking the C-terminal polyubiquitin binding domain (ΔBUZ), or lacking the SE14 domain that mediates protein–protein interactions (ΔSE14) (Supplementary Fig. 1b). Neither polyubiquitin binding nor HDAC6 catalytic activity was required for HDAC6 binding to tau, however, removal of the C-terminal domain (1–840) or just deletion of the minimal SE14 domain (ΔSE14) significantly reduced tau binding without affecting total tau levels (Fig. 1b, c and Supplementary Fig. 1d). These findings suggest that HDAC6 binding to tau is ubiquitin-independent and mediated, in part, by the SE14 domain.

We also generated a complimentary series of tau deletion constructs to dissect the regions of tau that are essential for HDAC6 binding (Supplementary Fig. 1a). Full-length WT human tau (2N4R) contains two N-terminal repeats and all four MTBR domains (R1–R4). Repeats R2 and R3 harbor two homologous hexapeptide motifs known as PHF6* ($^{275}$VQIINK$^{280}$) and PHF6 ($^{306}$VQIVYK$^{311}$), which are implicated in tau aggregation[32,33]. Tau constructs lacking individual repeats or all four repeats (ΔR1–4) were analyzed by co-IP assays with WT HDAC6 (Fig. 1d). While deletion of R1 and R4 did not impair HDAC6 binding, deletion of R2, and most prominently R3, significantly reduced binding when compared to full-length WT tau (Fig. 1d, e). Removal of the entire MTBR domain (ΔR1–4) nearly eliminated tau-HDAC6 binding (Fig. 1d, e).

Given that six alternatively spliced tau isoforms are expressed in the adult human brain, we next asked whether HDAC6 preferentially bound 4R-tau isoforms compared to 3R-tau isoforms that lack R2 due to exon 10 alternative splicing of the *MAPT* transcript. HDAC6 binding to 3R-tau isoforms (2N3R, 1N3R, and 0N3R) was slightly reduced when compared to the R2-containing 4R-tau isoforms (2N4R, 1N4R, and 0N4R) (Fig. 1f, g). The presence or absence of tau N-terminal inserts did not appreciably alter tau–HDAC6 binding, further implicating the MTBR as the critical determinant of the tau–HDAC6 interaction. Additionally, a panel of frontotemporal dementia (FTD) linked tau mutations (Supplementary Fig. 1a), many of which cluster in the R2 and R3 regions, showed a range of binding with some mutants showing increased HDAC6 binding (e.g., P301L and S320F) while others showed reduced HDAC6 binding (e.g., ΔK280 and L315R) (Fig. 1h, i).

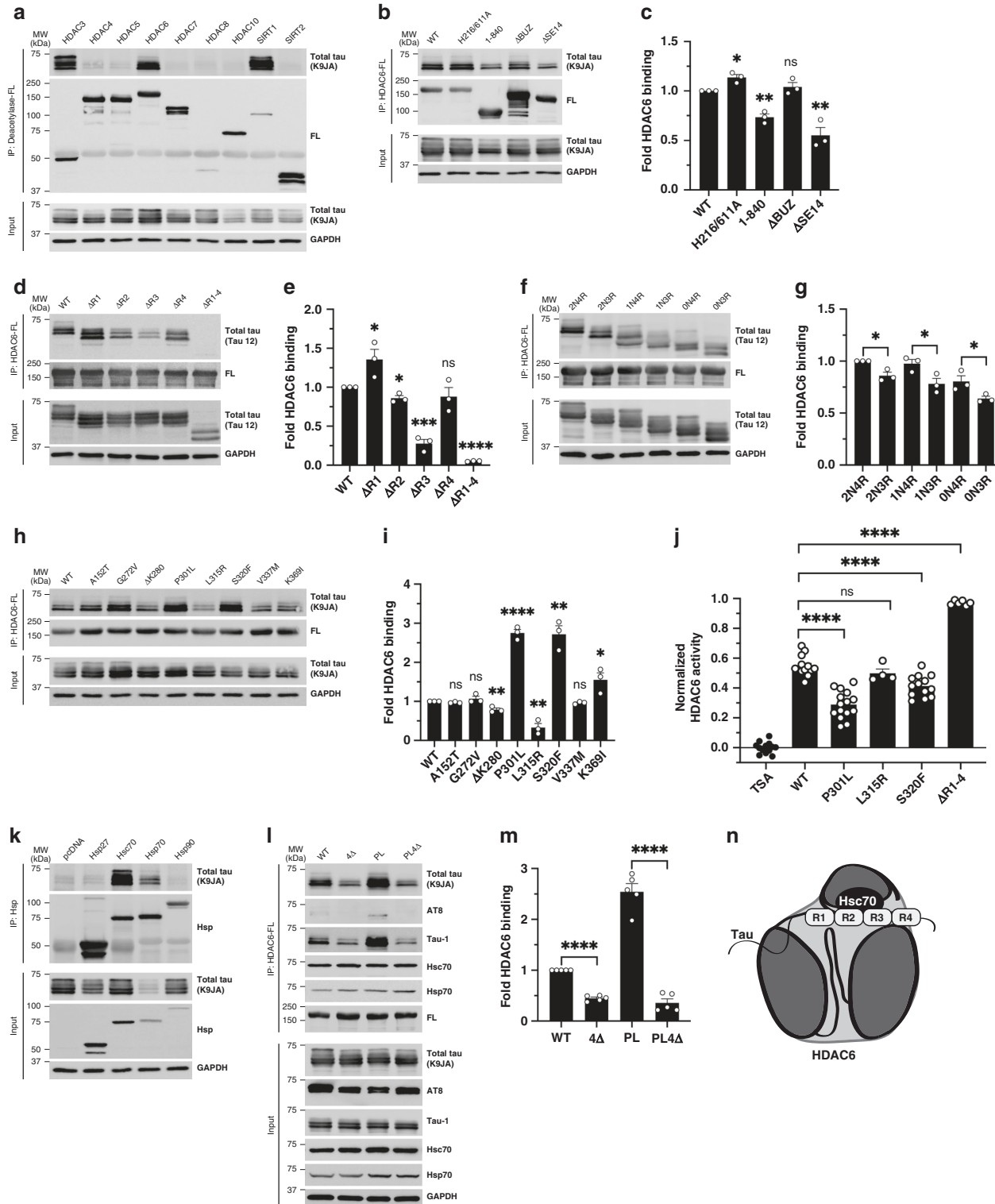

To further examine the association of tau with HDAC6, we performed in vitro HDAC6 deacetylase assays reconstituted with recombinant purified tau and HDAC6 proteins as well as a fluorescent HDAC reporter. The P301L and S320F tau mutants, which show enhanced HDAC6 binding, were also more effective at sequestering and impairing HDAC6 activity while the L315R mutant, which showed reduced HDAC6 binding, did not appreciably inhibit HDAC6 activity (Fig. 1j). By extending our analysis to other HDACs, we found that the enhanced inhibitory activity of P301L was specific to HDAC6, when compared to

HDAC1 or HDAC3 (Supplementary Fig. 1e). Furthermore, the HDAC6-binding deficient ΔR1–4 tau mutant (which lacks the MTBR interacting domain), fully restored HDAC6 activity but did not restore HDAC1 or HDAC3 activity (Supplementary Fig. 1e). Thus, binding of the tau R2/R3 aggregate-prone motifs to HDAC6 is sufficient to impair HDAC6 activity, an effect that is modulated by the presence of disease-linked familial tau mutations.

Heat shock proteins (Hsps) including Hsp70 family members interact with tau via the R2 and R3 motifs in the MTBR[34].

**Fig. 1 An HDAC6-chaperone complex targets aberrant tau species. a** Co-IP assay evaluating the binding of a panel of human deacetylases to tau in HEK-293A cells by FLAG (FL) pull-down followed by immunoblotting with a total tau (K9JA) antibody. WT (wild-type) tau preferentially interacts with HDAC3, HDAC6, and SIRT1 as indicated. **b** Co-IP assay to determine HDAC6 interacting domains. WT HDAC6, H216/611A (catalytically dead HDAC6 harboring both H216/H611A mutations), 1–840 (lacking 841–1215), ΔBUZ (lacking 1045–1215), and ΔSE14 (lacking 884–1022) were evaluated. C-terminal mutants (1–840 and ΔSE14) exhibited decreased binding to tau, as quantified in **c**. Values are means ± SEM of the ratio of IP total tau to input total tau (background subtracted and normalized to WT) with unpaired $t$-test statistical analysis for each HDAC6 mutant compared to WT HDAC6 (CD, $*p = 0.0107$; 1–840, $**p = 0.0013$; ΔBUZ, ns; ΔSE14, $**p = 0.0048$). **d** Co-IP assay using tau mutants lacking individual MTBR repeats or the entire MTBR (R1, Q244-K274; R2, V275-S305; R3, V306-Q336; R4, V337-E372; R1-4, Q244-E372). Removal of R2, R3, or complete removal of the MTBR (R1–4) reduced HDAC6 binding. The binding of MTBR mutants to HDAC6 were quantified in **e** (ΔR1, $*p = 0.0498$; ΔR2, $*p = 0.0125$; ΔR3, $***p = 0.0002$; ΔR4, ns; ΔR1–4, $****p = <0.0001$). **f** Co-IP assay assessing HDAC6 binding to tau isoforms varying in the number of N-terminal inserts (0N, 1N, or 2N) or MTBR domains (3R or 4R). 3R-tau (0N3R, 1N3R, and 2N3R) and 4R-tau (0N4R, 1N4R, and 2N4R) binding was quantified in **g** (2N4R vs. 2N3R, $*p = 0.0125$; 1N4R vs. 1N3R, $*p = 0.0435$; 0N4R vs. 0N3R, $*p = 0.0499$). **h** Co-IP assay assessing HDAC6 binding to FTD disease-causing tau mutations was quantified in **i** (A125T, ns; G272V, ns; ΔK280, $**p = 0.0069$; P301L, $****p < 0.0001$; L315R, $**p = 0.0029$; S320F, $**p = 0.0013$; V337M, ns; K369I, $*p = 0.0384$). **j** HDAC6 activity assays were used to evaluate the ability of full-length recombinant WT, P301L, L315R, S320F, and ΔR1–4 tau proteins to inhibit HDAC6 activity, which were normalized to the TSA control. Values are means ± SEM. **k** Co-IP assay to evaluate tau binding to ectopically over-expressed heat shock proteins (Hsps) including Hsp27, Hsc70, Hsp70, and Hsp90. Only Hsc70 and Hsp70 bound robustly to tau. **l** Co-IP assay to evaluate HDAC6 binding to WT and P301L (PL) compared to their Hsc/Hsp70 binding-deficient mutants (4Δ and PL4Δ; ΔI277/I278/V308/V309). HDAC6 binding was decreased in the presence of chaperone binding-deficient tau mutations, as quantified in **m. n** Schematic of the HDAC6-chaperone complex in association with tau. Statistical tests: $p$ value determined by two-sided unpaired $t$-test (**c, e, g, i, j, m**) from $n = 3$ (**e, g, i**), $n = 4$ (**j**), and $n = 5$ (**m**) biologically independent experiments. Representative co-IP assays (**a, k**) are shown from $n = 3$ biologically independent experiments. Error bars represent means ± SEM (**c, e, g, i, j, m**). $*p < 0.05$; $**p < 0.01$; $***p < 0.001$; $****p < 0.0001$. Source data are provided as a Source Data file.

Similarly, HDAC6 interacts with Hsps (e.g., Hsp70 and Hsp90) as part of a PQC pathway that responds to misfolded and cytotoxic protein aggregates[35–37]. Given the shared interaction with Hsps, we asked whether tau might bind HDAC6 via a bridged chaperone intermediate by evaluating a tripartite tau–Hsp–HDAC6 complex. Co-IP assays with individual Hsps showed that tau exhibited the strongest binding to Hsp70 and highly related Hsc70, rather than other Hsp family members including Hsp27 and Hsp90 (Fig. 1k). We note that Hsp70, but not Hsc70, enhanced tau clearance based on the reduced levels of total tau observed in the presence of Hsp70 (Fig. 1k, see total tau input). This finding is consistent with previous reports that Hsp70 facilitates tau degradation[38]. Further supporting a HDAC6–Hsp–tau complex, deletion of the SE14 domain in HDAC6 similarly reduced the binding of HDAC6 to Hsp70 and Hsc70 (Supplementary Fig. 1f).

Next, we generated tau mutants that were unable to associate with Hsc70 by deleting four hydrophobic residues in R2 (I277/I278) and R3 (I308/V309) known to mediate the tau–Hsc70 interaction[34], thereby generating an Hsc70-binding deficient (4Δ) mutant (Supplementary Fig. 1a). By abolishing the tau–Hsc70 association in the context of full-length WT tau (4Δ), and more prominently in the context of P301L tau that showed increased binding to HDAC6 (PL4Δ), we observed a dramatic reduction of tau–HDAC6 binding (Fig. 1l, m and Supplementary Fig. 1g, h). We note that phosphorylated tau (AT8 epitope) showed minimal association with HDAC6 when compared to dephosphorylated tau (Tau-1 epitope)[39] that bound HDAC6 similar to total tau, suggesting HDAC6 preferentially targets a less phosphorylated tau species (Fig. 1l).

We also used the acetyltransferase CBP to drive excessive tau acetylation and observed impaired tau binding to both Hsc70 and Hsp70, whereas the presence of overexpressed HDAC6 conversely promoted tau binding to Hsps (Supplementary Fig. 1i), suggesting abnormally acetylated tau is sufficient to abrogate Hsp70/Hsc70 docking onto tau. Overall, our data suggest that the HDAC6–Hsp complex targets the aggregate-prone tau MTBR as part of a cellular surveillance mechanism (Fig. 1n).

**HDAC6 localizes to sites of aberrant tau accumulation.** To examine tau and HDAC6 localization and their potential coaggregation in neurons, dissociated primary cortical neurons from tau knockout mice (Tau KO) were transfected with P301L Tau-GFP and HDAC6-FL expression plasmids. Abundant cytoplasmic colocalization between tau and HDAC6 was observed both in the soma and along neuritic processes (Fig. 2a and Supplementary Fig. 2a). We also evaluated the localization of endogenous tau and HDAC6 in cortical neurons derived from WT and P301S tau transgenic mice (PS19). Tau and HDAC6 were cytoplasmic in WT neurons (Fig. 2b and Supplementary Fig. 2b, see top row), but colocalized within discrete foci, or swellings, that were deposited along the processes of PS19 neurons (Fig. 2b and Supplementary Fig. 2b, see bottom row). These Tau-1-immunoreactive swellings, or varicosities, have been observed in degenerating axons, in response to neuronal damage, and during neuroinflammation[40].

To further evaluate whether HDAC6 is linked to tau pathology in human brain, control and AD brain tissues were homogenized, fractionated, and examined by immunoblotting. In AD brains, soluble HDAC6 levels were low and accumulated in the insoluble aggregated fractions, migrating as higher molecular weight smears of ~150–300 kDa (Fig. 2c, see insoluble HDAC6). Aggregated HDAC6 tracked with tau NFT pathology, which was invariably present in all AD brains but not controls (Fig. 2c, see insoluble p-S202). We also observed reduced soluble acetylated tubulin, which increased in the insoluble fractions of AD cases, indicating compromised HDAC6 function (Fig. 2c, see ac-tubulin). Quantification of immunoblotting confirmed a solubility shift and inactivation of HDAC6 (Fig. 2d, e). Finally, HDAC6 associated with Thioflavin S (ThS)-positive neuritic plaques in human AD cortex by confocal imaging (Fig. 2f). The plaque-associated pool of HDAC6 co-localized with total tau (Fig. 2f, TAU-5, see bottom row) and conformational MC1-positive tau inclusions (Fig. 2f, see top row), an early indicator of misfolded tau species. These observations suggest that HDAC6 coaggregates with tau and undergoes recruitment into AD lesions.

**Pathological tau is acetylated at HDAC6 target sites.** We next sought to identify the major tau PTMs that emerge as a result of soluble HDAC6 depletion in AD brain. Human AD cortex was biochemically extracted and soluble tau proteins corresponding to all six tau isoforms were immunoprecipitated (using a combination of N-terminal T14 and C-terminal T46 monoclonal tau antibodies) and analyzed by liquid chromatography and mass

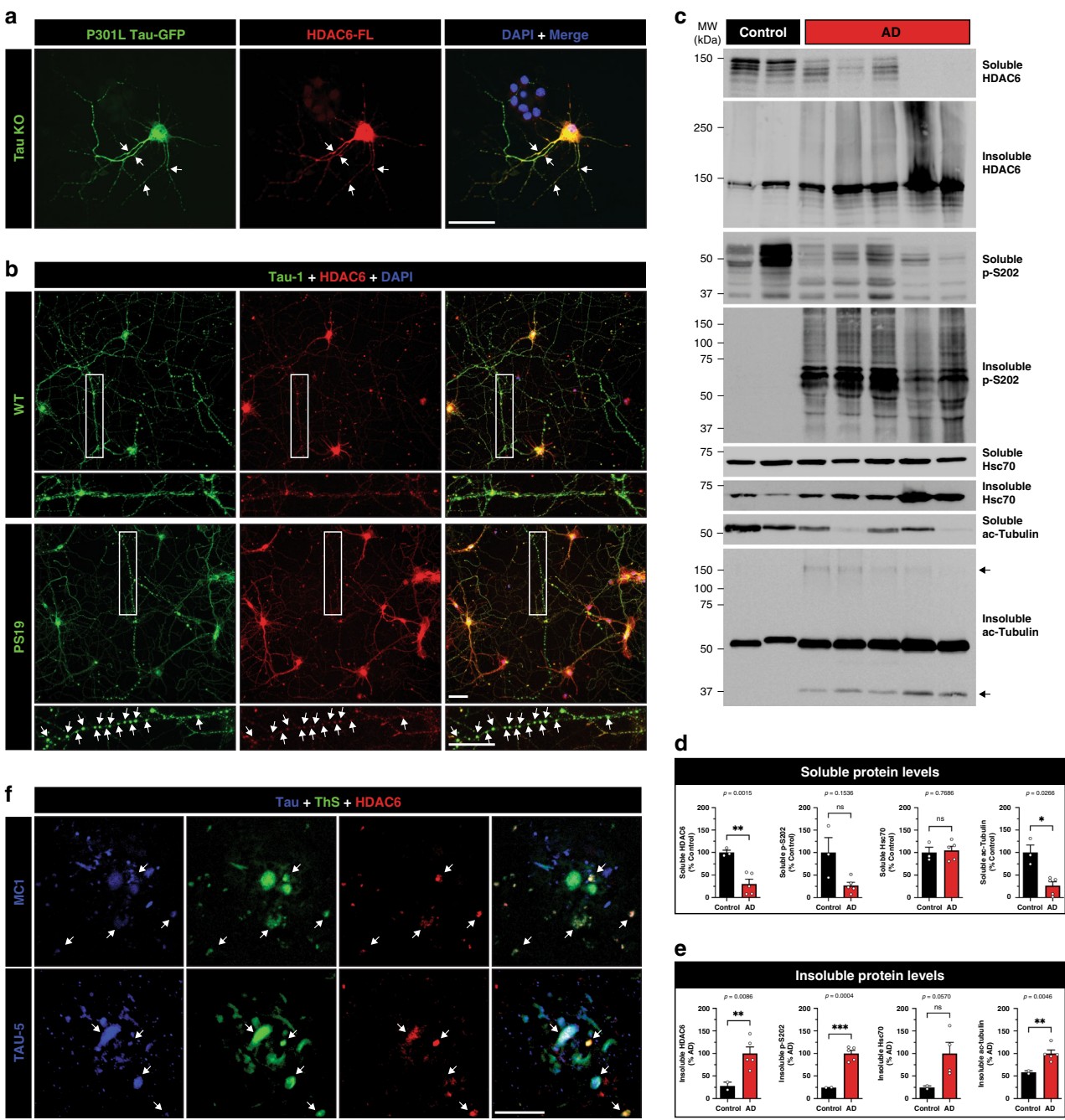

**Fig. 2 HDAC6 co-aggregates with tau and accumulates in pathological tau lesions. a** Primary cortical neurons from mouse embryos (E16) of tau knockout (KO) mice were cultured for 10 days in vitro (10 DIV) and transfected with P301L Tau-GFP (green) and HDAC6-FLAG (FL) (red) expression constructs followed by imaging. Arrows indicate regions of extensive co-localization along processes. Scale bar = 50 μm. **b** Primary cortical neurons (E16) from WT (top) and PS19 tau transgenic (bottom) mice were analyzed by immunofluorescence (IF), detecting co-localized focal swellings of tau (green), endogenous HDAC6 (red), and the nuclei marked by DAPI (blue). The white rectangle highlights a magnified insert of a neuronal process harboring tau-positive foci. Scale bar = 50 μm. **c** Biochemical fractionation of soluble high-salt fractions and insoluble urea fractions were extracted from frontal cortex of control and AD brains followed by immunoblotting to detect HDAC6, tau, Hsc70, and acetylated tubulin. The arrows highlight acetylated tubulin fragments ~37 kDa and higher molecular weight species ~150 kDa. **d, e** Immunoblotting results were quantified by densitometry. **f** AD cortical sections were analyzed by double-labeling IF using either conformational tau (MC1) or total tau (TAU-5) antibodies (blue), combined with Thioflavin S (ThS) (green), and HDAC6 (red) to evaluate localization in proximity to AD neuritic plaques. White arrows highlight regions of tau and HDAC6 immunoreactivity that are associated with ThS positive plaques. Scale bar = 50 μm. Statistical tests: p value determined by two-sided unpaired t-test with Welch's correction from n = 5 biologically independent samples (**d, e**). Representative neurons are shown from n = 3 biologically independent experiments (**a, b**) and n = 3 independent AD brains (**f**). Error bars represent means ± SEM and are plotted relative to control (**d**) or relative to AD (**e**). *p < 0.05, **p < 0.01, ***p < 0.001. Source data are provided as a Source Data file.

**Table 1 List of tau modifications identified from human AD brains.**

| PTM-Residue | Peptide | (−10logp) | ppm |
|---|---|---|---|
| p-S46 | [45]ES**P**LQTPTEDGSEEPGSETSDAK[67] | 15.13 | 1.0 |
| p-T50 | [45]ESPLQ**T**PTEDGSEEPGSETSDAK[67] | 43.49 | −1.2 |
| p-T175 | [171]IPAK**T**PPAPK[180] | 53.71 | −0.3 |
| p-T181 | [175]TPPAPK**T**PPSSGEPPK[190] | 60.80 | −0.6 |
| p-S184 | [175]TPPAPKTPP**S**SGEPPK[180] | 21.79 | −3.1 |
| p-S198 | [195]SGY**S**SPGSPGTPGSR[209] | 53.54 | 0.2 |
| p-S199 | [195]SGYS**S**PGSPGTPGSR[209] | 53.91 | −1.1 |
| p-S202 | [195]SGYSSPG**S**PGTPGSR[209] | 44.83 | 0.1 |
| p-S210 | [210]**S**RTPSLPTPPTR[221] | 19.07 | 0.2 |
| p-T212 | [212]**T**PSLPTPPTREPK[224] | 34.30 | −0.3 |
| p-S214 | [210]SRTP**S**LPTPPTR[221] | 52.35 | −0.5 |
| p-T217 | [212]TPSLP**T**PPTREPK[224] | 42.92 | 0.4 |
| p-T231 | [226]VAVVR**T**PPKSPSSAK[240] | 40.57 | 0.7 |
| p-S235 | [225]KVAVVRTPPK**S**PSSAK[240] | 28.16 | 0.1 |
| p-S237 | [226]VAVVRTPPKSP**S**SAK[240] | 43.63 | 0.3 |
| p-S238 | [226]VAVVRTPPKSPS**S**AK[240] | 34.65 | 0.2 |
| ac-K259 | [258]S**K**IGSTENLK[267] | 15.79 | −0.1 |
| p-S262 | [260]IG**S**TENLKHQPGGGK[274] | 29.82 | −0.5 |
| ac-K311 | [306]VQIVY**K**PVDLSK[317] | 39.04 | 0.0 |
| ac-K385 | [384]A**K**TDHGAEIVYK[395] | 46.13 | 0.6 |
| p-S396 | [386]TDHGAEIVYK**S**PVVSGDTSPR[406] | 48.12 | 0.0 |
| p-T403 | [386]TDHGAEIVYKSPVVSGD**T**SPR[406] | 54.50 | 0.1 |
| p-S404 | [396]SPVVSGDT**S**PR[406] | 57.89 | −0.7 |

Tau proteins were isolated from AD brain extracts and purified for LC-MS/MS analysis to identify sites of tau phosphorylation and acetylation. The residue number, peptide sequence (bold denotes the modified residue), and significance values are depicted. The tau sequences listed are based on full-length human 2N4R tau (441 residues). The protein confidence score p is represented as −Log p. Peptide mass accuracy is represented as parts per million (ppm).

spectrometry (LC/MS–MS). As shown in Table 1, many tau PTMs were identified in AD brains including phosphorylation at N-terminal, mid-domain, and C-terminal residues. Immunoprecipitated soluble tau contained three acetylation sites at tau residues K259, K311, and K385 (Table 1). Acetylation at residue K311 was of primary interest for the following reasons. K311 resides in R3 of the MTBR, a region that physically binds HDAC6 (Fig. 1d, e). In addition, R3 contains the PHF6 hexapeptide motif ([306]VQIVYK[311]) known to mediate tau aggregation[33], which is homologous to the R2-containing PHF6* motif ([275]VQIINK[280]) that is subject to acetylation at the corresponding K280 residue leading to tau dissociation from MTs and enhanced aggregation[1,41].

We hypothesized that tau acetylation at the K311 residue might contribute to tau aggregation, given its central location in the PHF6 motif that mediates tau aggregation. To test this possibility in vitro, we purified recombinant tau proteins (K18 tau containing only the MTBR) and inserted either acetylation-mimic (K → Q) or deacetylation-mimic (K → R) mutations at K311 for subsequent aggregation and oligomerization assays (Supplementary Fig. 3). K311Q showed accelerated aggregation kinetics and completely transitioned from the soluble supernatant fraction (Sup) to the aggregated pellet fraction (Pel) by 1 h, an effect that was comparable to the FTD disease-linked P301L mutant (Supplementary Fig. 3a). Aggregated K311Q was also characterized by dimers and trimers ~30 and 45 kDa, suggestive of higher order multimeric tau species. In contrast, the K311R deacetylation-mimic showed delayed aggregation kinetics and did not achieve complete aggregation until 8 h similar to WT tau. These results were also quantified by Thioflavin T (ThT) fluorescence, which showed a rapid increase in K311Q aggregation leading to ~3-fold increase in ThT-positive fibrils by 8 h compared to WT or K311R mutant tau (Supplementary Fig. 3b).

Using dynamic light scattering, we further confirmed the presence of tau oligomers (Supplementary Fig. 3c). Only K311Q and P301L formed higher order structures as early as 15 min, which gradually progressed in intensity until 1 h, at which point tau fully aggregated, in contrast to WT or K311R mutant. Therefore, mimicking acetylation at only a single residue (K311), in the absence of other tau modifications, was sufficient to accelerate tau oligomerization and aggregation.

To begin characterizing tau acetylation at residue K311 in the brain, we generated acetylation site-specific K311 polyclonal antibodies (hereafter referred to as ac-K311). One of the four purified antibodies (ac-K311, #5089) showed high specificity and sensitivity for acetylated tau, but not unmodified tau (Supplementary Fig. 4a, b). The specificity was demonstrated by minimal detection of an acetylation-deficient K311R mutation, suggesting the immunoreactivity is directed primarily towards acetylation of the K311 residue (Supplementary Fig. 4b). Ac-K311 did not distinguish among tau isoforms in vitro, as CBP-mediated acetylation of all six tau isoforms was observed in cultured HEK-293A cells (Supplementary Fig. 4c). The highest specificity clone (#5089) was subsequently employed for the analysis of AD and related tauopathy cases.

Immunoblotting of AD brain homogenates with ac-K311 detected robust insoluble aggregated high molecular weight tau species, similar to mature tau pathology detected with total tau antibodies (Fig. 3a, b). Consistent with ac-K311 labeling tau pathology, immunohistochemistry (IHC) analysis detected widespread neuropil thread and NFT pathology in all AD brains analyzed, but not healthy control brains (Fig. 3c and Supplementary Fig. 5a). While 3R-tau predominant Pick's disease (PiD) was also strongly immunoreactive (Fig. 3c, see ac-K311 positive Pick bodies), surprisingly, 4R-tau predominant corticobasal degeneration (CBD) and progressive supranuclear palsy (PSP) showed minimal ac-K311 immunoreactivity (Fig. 3c, see ac-K311 negative CBD and PSP cases), despite the presence of phosphorylated tau inclusions in all tauopathy cases analyzed (Supplementary Fig. 5b). The lack of ac-K311 pathology in CBD and PSP brain was unexpected since residue K311 is present in all six tau isoforms (both 3R-tau and 4R-tau) that form inclusions in AD (mixed 3R/4R tau pathology), PiD (3R tau pathology), and CBD/PSP (4R tau pathology). To confirm these findings, double-labeling immunofluorescence showed colocalization of ac-K31l with 3R-tau inclusions in AD and PiD (Fig. 3d). In AD brains, ac-K311 labeled 3R-tau pathology more robustly than 4R-tau pathology based on prominent colocalization with the 3R-tau specific antibody RD3 (Fig. 3d, compare top row AD–RD3 with AD–RD4). Similar to the IHC analysis, neuronal and glial pathology in CBD (astrocytic plaques) and PSP (globose tangles and tufted astrocytes) were largely undetected by ac-K311 (Fig. 3d, see negative ac-K311 immunoreactivity in CBD and PSP cases). To further confirm disease-specificity, we screened an additional 42 CBD/PSP cases and found only four cases with rare ac-K311 immunoreactivity that was in the form of AD-like neuritic plaques and neuronal tangles that were both RD3 and RD4 positive, suggesting ac-K311 immunoreactivity predominantly occurs in 3R-tauopathies (Supplementary Table 1).

These observations were further evaluated by immunoblotting purified PHF-tau preparations isolated from a panel of tauopathy brains[42]. Detection of ac-K311 tau bands were observed in AD, and more prominently in PiD, migrating at ~60–65 kDa (bands labeled #3 and #5), but were not present in CBD or PSP brains (Fig. 3e). The ac-K311 bands were immunoreactive with the RD3 antibody, supporting the notion that ac-K311 preferentially detects 3R-tau. Consistent with the 3R-tau specificity, we evaluated ac-K311 immunoreactivity in cortical sections from a human FTLD-tau P301L case harboring predominantly 4R-tau

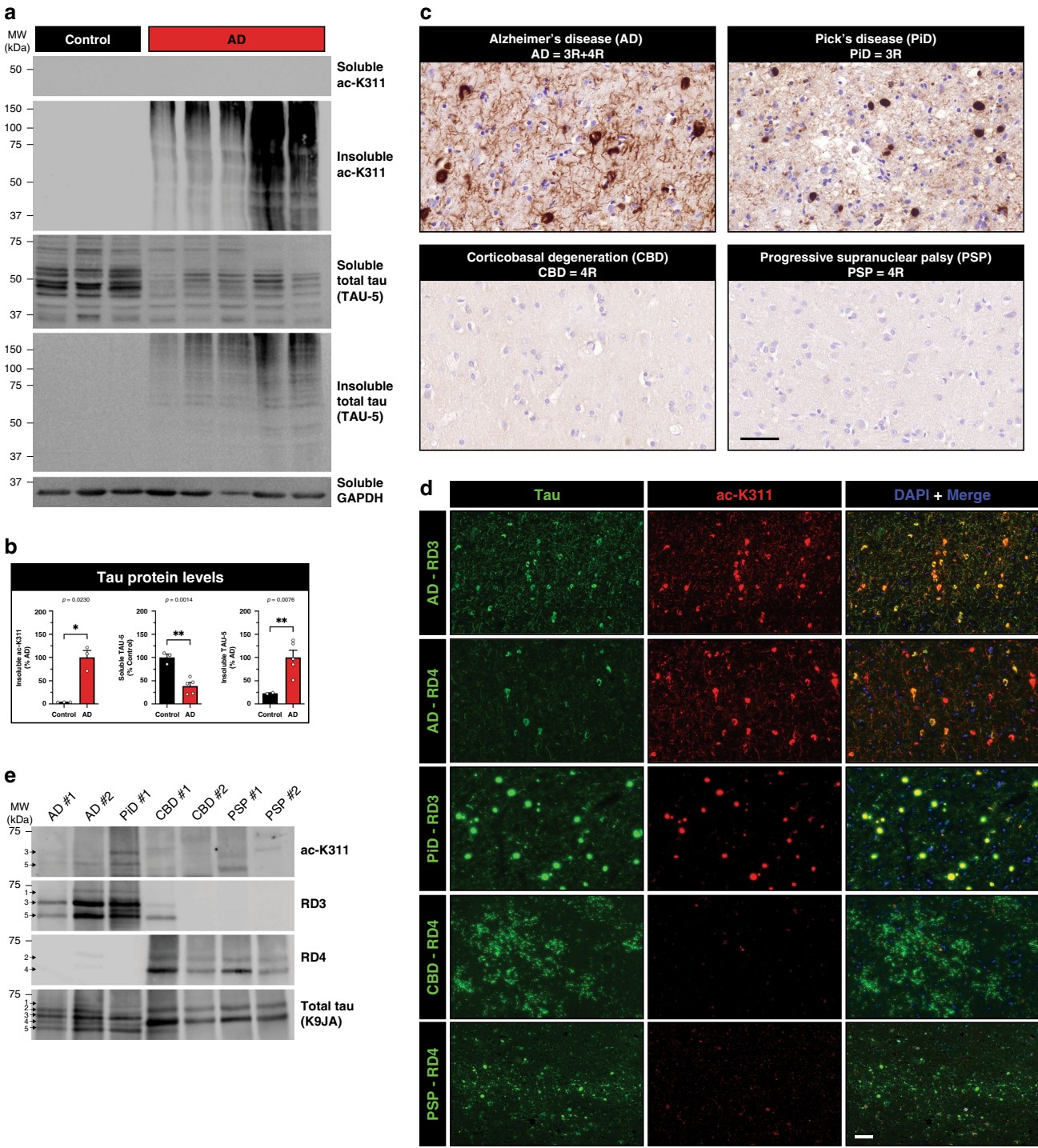

**Fig. 3 AD and related tauopathies present with distinct tau acetylation profiles. a** Immunoblotting of control and AD brains homogenates using antibodies to detect ac-K311, total tau (TAU-5), and GAPDH. **b** Immunoblots from **a** were quantified and normalized to control (soluble) or AD (insoluble) samples. **c** Immunohistochemistry (IHC) analysis using ac-K311 (#5089) in a panel of 3R and 4R tauopathies: Alzheimer's disease (AD), Pick's disease (PiD), corticobasal degeneration (CBD), and progressive supranuclear palsy (PSP). Only 3R-tauopathies (AD and PiD) demonstrated ac-K311-immunoreactive pathological lesions. Scale bar = 50 μm. **d** Double-labeling IF to mark ac-K311 (red) in combination with tau-isoform specific antibodies. The green channel represents 4R-tau (RD4) in AD, CBD, and PSP or 3R-tau (RD3) in AD and PiD. Ac-K311 colocalized strongly to 3R-tau pathology in AD and PiD. Scale bar = 50 μm. **e** Immunoblotting of isolated PHF-tau from the indicated tauopathies probed with ac-K311, 3R-tau (RD3), 4R-tau (RD4), and total tau (K9JA) antibodies. The arrows highlighting tau bands 1–5 represent distinctly migrating tau bands that are resolved by immunoblotting. Statistical tests: $p$ value determined by unpaired $t$-test with Welch's correction from $n = 5$ independent samples (**b**). Representative images are shown from $n = 10$ (**c**, **d**) and $n = 2$ (**e**) independent samples. Error bars represent means ± SEM. *$p < 0.05$ and **$p < 0.01$. Source data are provided as a Source Data file.

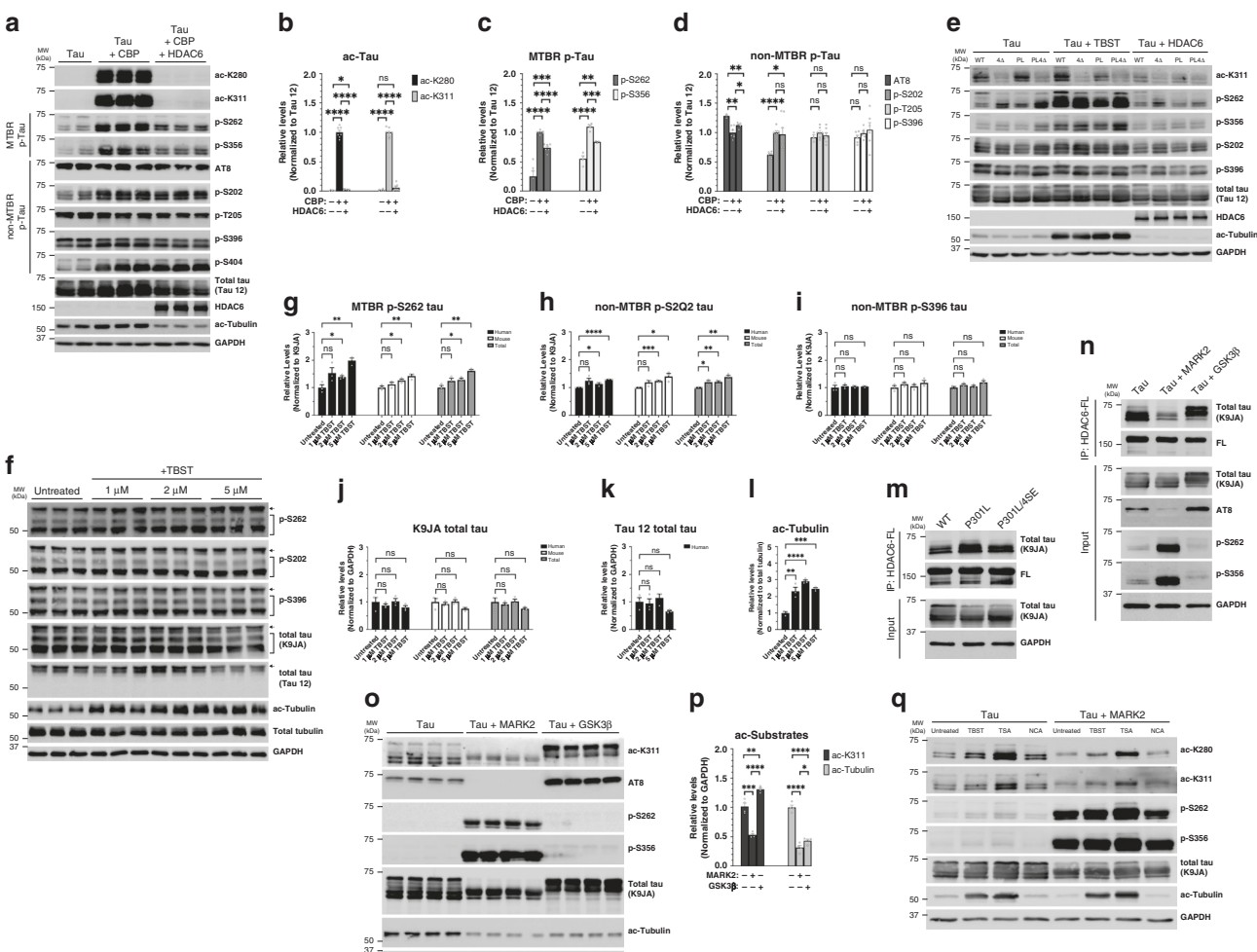

**Fig. 4 The targeting of tau by HDAC6 is regulated by tau kinases. a** Immunoblotting of HEK-293A cells cotransfected for 48 h with WT tau, the acetyltransferase CBP, and WT HDAC6 (where indicated) using acetylation or phosphorylation specific tau antibodies. **b–d** The relative levels of modified tau within the MTBR (ac-280, ac-K311, p-S262, and p-S356) or outside the MTBR (AT8, p-S202, p-T205, and p-S396) were quantified by densitometry of tau PTMs normalized to total tau (Tau 12). **e** Cells expressing the indicated tau constructs (WT, 4Δ, PL, and PL4Δ) were treated with tubastatin A (TBST) to inhibit HDAC6 or cotransfected with WT HDAC6 to suppress tau acetylation followed by immunoblotting with acetylation or phosphorylation specific tau antibodies as in **a**. **f–l** Primary cortical neurons (E16) from PS19 derived tau-transgenic mice cultured for ten DIV were treated overnight with TBST at a concentration of 1–5 μM and analyzed by immunoblotting followed by quantification by densitometry of tau PTMs normalized to human, mouse, or total tau (K9JA). The arrows in **f** highlight overexpressed human tau migrating at ~65 kDa compared to mouse tau at ~50 kDa (see also Supplementary Fig. 6c to distinguish between human and mouse tau). **m** Co-IP assay evaluating HDAC6 binding to WT tau, tau-P301L, and tau-P301L/4SE mutant containing S → E substitutions at all four MARK2-targeted phosphorylation sites (S262/S290/S324/S356E). Despite P301L tau showing greater binding to HDAC6, the addition of the MARK2 phosphorylation-mimic mutations reduced HDAC6 binding. **n** Co-IP assay evaluating tau-P301L mutant tau binding to HDAC6 in the presence of MARK2 or GSK-3β kinases. **o** HEK-293A cells co-transfected with WT tau, MARK2, or GSK-3β (where indicated) were analyzed by immunoblotting using acetylation or phosphorylation specific tau antibodies followed by quantification of ac-K311 and ac-Tubulin levels normalized to GAPDH levels in **p**. **q** HEK-293A cells co-transfected with WT and MARK2 were treated with a panel of HDAC inhibitors including tubastatin A (TBST, HDAC6 inhibitor, 5 μM), trichostatin A (TSA, class I/II HDAC inhibitor, 1 μM) or nicotinamide (NCA, class III HDAC inhibitor, 5 mM) and analyzed as described in **o**. Statistical tests: p value determined by two-sided unpaired t-test from n = 6 (**b–d**), n = 3 (**g–l**), and n = 4 (**p**) biologically independent experiments. Error bars represent means ± SEM (**b–d, g–l, p**). *p < 0.05; **p < 0.01; ***p < 0.001; ****p < 0.0001. Source data are provided as a Source Data file.

pathology, which also showed minimal ac-K311 immunoreactivity (Supplementary Fig. 5c). The lack of ac-K311 immunoreactivity in 4R tauopathies compared to 3R tauopathies implies the existence of a unique tau profile, tau conformation, and/or tau strain that distinguishes 3R from 4R tauopathies. While ac-K311 showed specificity for 3R-tau in diseased human tauopathy brains, we note that this was not observed in vitro, as both 3R-tau and 4R-tau isoforms showed identical patterns of acetylation mediated by CBP in cultured cells (Supplementary Fig. 4c). Therefore, all tau isoforms have the potential to be acetylated at

K311 in vitro, however in human tauopathy brains, K311 acetylation emerges in a disease and tau isoform-specific pattern.

We next examined whether HDAC6 is sufficient to target and repair aberrant tau PTMs. CBP-acetylated tau was potently deacetylated by HDAC6 using site-specific ac-K280 and ac-K311 antibodies (Fig. 4a, b). Further analysis with a panel of phosphorylated tau antibodies showed that CBP-mediated tau acetylation also led to increased tau phosphorylation at S262 and S356 (MARK2-dependent target KXGS motifs) as well as S202 and S404, but not other AD-relevant epitopes in the proline-rich

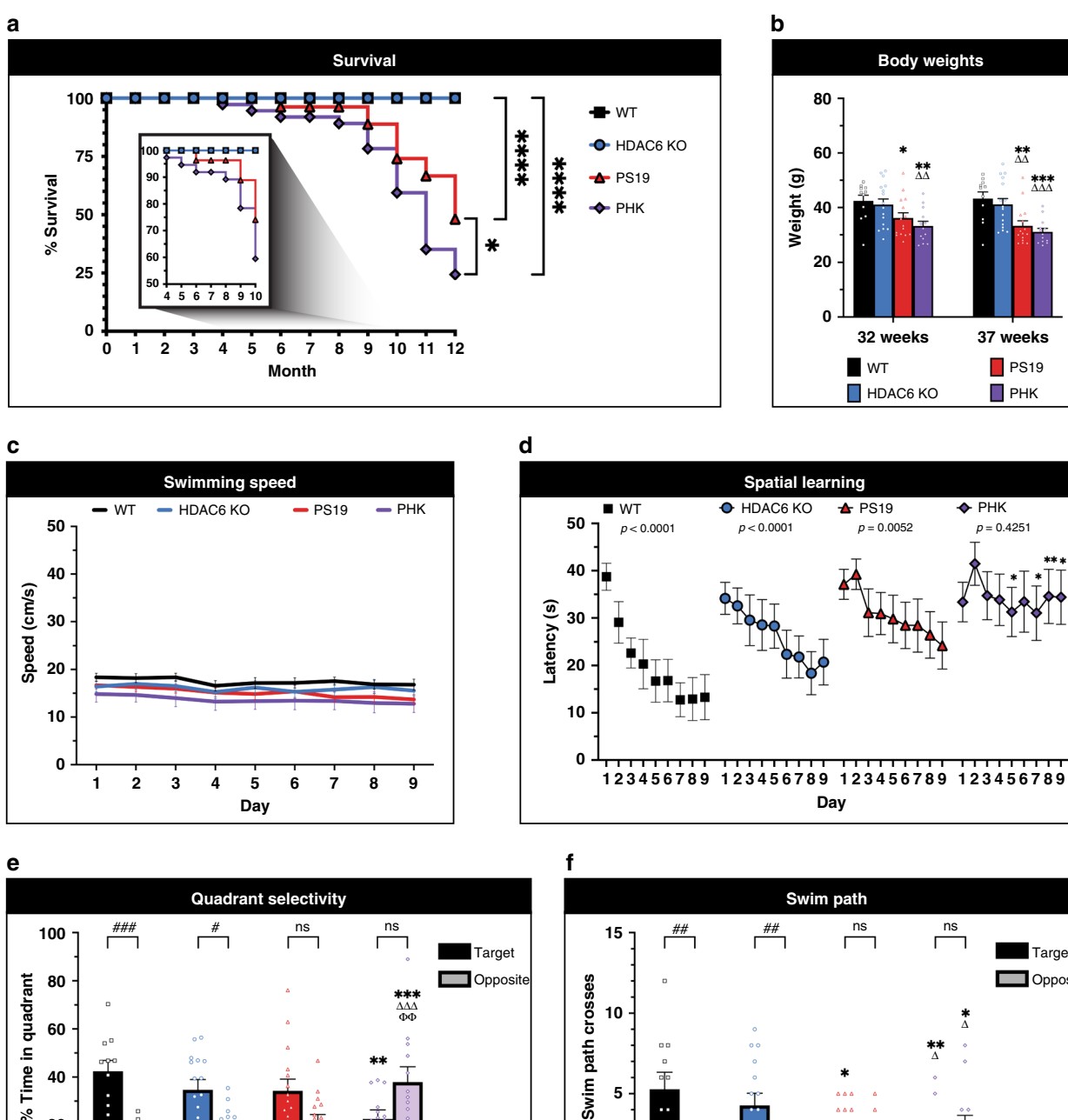

(AT8 and T205) or C-terminal (S396) regions (Fig. 4a–d). Surprisingly, HDAC6 not only reduced acetylation but also reduced tau phosphorylation exclusively at MARK2 target sites S262 and S356 (Fig. 4a–d). To evaluate which HDAC6 catalytic domain (CD1 or CD2) is responsible for tau deacetylation, we analyzed mutations in CD1 (H216A) or CD2 (H611A) and found that inactivation of CD2 is sufficient to restore CBP-mediated tau acetylation, which suggests that HDAC6 catalytic activity towards tau substrates resides in the CD2 domain (Supplementary Fig. 6a).

We examined whether HDAC6 inhibition conversely promoted S262/S356 phosphorylation within the MTBR. Indeed, either abrogation of the tau–HDAC6 interaction using the 4Δ mutations (which reduces tau–HDAC6 binding) or pharmacological inhibition

of HDAC6 activity using tubastatin A (TBST), were sufficient to increase phosphorylation in the MTBR with less prominent effects on nonMTBR tau phosphorylation sites (e.g., S202 and S396) (Fig. 4e). Furthermore, ectopic overexpression of HDAC6 reduced S262/S356 phosphorylation, even in the presence of the PL4Δ mutant that was susceptible to hyperphosphorylation (Fig. 4e). Surprisingly, tau mutants that are HDAC6 binding-deficient (either 4Δ or PL4Δ) showed less ac-K311 signal (Fig. 4e and Supplementary Fig. 6b). The reduced acetylation may potentially be due to other unrelated HDACs targeting tau, since trichostatin A (TSA), a pan-HDAC inhibitor, restored the acetylation of the 4Δ mutant back to WT levels (Supplementary Fig. 6b). In contrast, inhibiting class III HDACs with nicotinamide (NCA) had no effect on ac-K311 levels (Supplementary Fig. 6b).

**Fig. 5 HDAC6 depletion in tauopathy mice reduces survival and accelerates cognitive decline. a** Survival analysis showed a significantly reduced survival rate in PS19 tau transgenic mice with HDAC6 deletion (PHK). Two-sided log-rank (Mantel-Cox) test with multiple comparisons was performed to assess significance among genotypes (WT, black squares, $n = 22$; HDAC6 KO, blue circles, $n = 42$; PS19, red triangles, $n = 27$; PHK, purple diamonds, $n = 37$) (*$p < 0.05$, ****$p < 0.0001$). **b** Decreased body weights in PS19 and PHK mice were observed at 32 weeks and 37 weeks of age (WT, $n = 11$; HDAC6 KO, $n = 15$; PS19, $n = 15$; PHK, $n = 13$). Comparisons among genotypes were as follows: *$p < 0.05$, **$p < 0.01$, ***$p < 0.001$, comparison to same measure in WT; $\Delta\Delta$, $p < 0.01$, $\Delta\Delta\Delta$, $p < 0.001$, comparison to HDAC6 KO. **c** No genotype differences were observed in average swimming speed across 9 days of training. **d** Acquisition of spatial learning using Morris water maze was evaluated for all genotypes (7–8-month old male mice). Data are means ± SEM from four trials per day. PHK mice showed impaired learning. $p$-values represent within-genotype repeated measures ANOVAs, effect of training day. WT, ****$p < 0.0001$; HDAC6 KO, ****$p < 0.0001$; PS19, **$p = 0.0052$; PHK, $p = 0.4251$. Asterisks represent comparisons to WT. **e** Quadrant preference data are means ± SEM from a 1-min probe trail without the platform following hidden platform training. Target indicates the quadrant where the platform had been placed during training, versus the opposite quadrant. Brackets signify within-genotype comparisons, effect of quadrant (#$p < 0.05$; ###$p < 0.001$; ns not significant). Comparisons among genotypes were as follows: **$p < 0.01$, ***$p < 0.001$, comparison to same measure in WT; $\Delta\Delta\Delta$, $p < 0.001$, comparison to HDAC6 KO; $\phi\phi$, $p < 0.01$ comparison to PS19. **f** Swim path crosses during a 1 min probe trial without the platform following hidden platform training. Target indicates the location where the platform had been placed during training, versus the corresponding location in the opposite quadrant. Within-genotype comparisons were used to determine effect of quadrant (##$p < 0.01$; ns not significant). Comparisons among genotypes were as follows: *$p < 0.05$, **$p < 0.01$, comparison to same measure in WT; $\Delta$, $p < 0.05$, comparison to HDAC6 KO. Statistical tests: $p$ value determined by repeated measure ANOVAs followed by Fisher's protected least-significant difference (PLSD) posthoc test from WT ($n = 11$), HDAC6 KO ($n = 16$), PS19 ($n = 15$), and PHK ($n = 13$) mice (**c–f**). All error bars throughout this figure represent means ± SEM (**b–f**). Source data are provided as a Source Data file.

To determine whether HDAC6 inhibition affects tau PTMs in neurons, we exposed PS19 cultured primary neurons to increasing concentrations (1–5 μM) of TBST (Fig. 4f). Similar to the HEK-293A cell-based results, TBST treatment led to an increase in S262 and S202 phosphorylation in PS19 neurons, but not S396 phosphorylation (Fig. 4f–i). We also analyzed both human transgenic tau and endogenous mouse tau to distinguish among the total tau species (Fig. 4j, k and Supplementary Fig. 6c). The human transgenic P301S mutant tau was more susceptible to S262 phosphorylation (~2-fold) in response to HDAC6 inhibition, as endogenous mouse tau was not as readily hyperphosphorylated (Fig. 4f, g). While HDAC6 inhibition increased the levels of acetylated tubulin as expected (Fig. 4l), HDAC6 inhibition is not sufficient on its own to generate acetylated tau at residues K280 or K311 in primary neurons lacking mature tau pathology. Active tau acetyltransferases (e.g., CBP/p300) may be required to promote acetylated tau levels to the extent observed in human tauopathies[1,43].

Given the potential interplay between tau phosphorylation and acetylation in the MTBR, we examined the impact of MARK2-mediated tau phosphorylation within the MTBR. We found that a tau phosphorylation-mimic containing four S → E substitutions at MARK2 phosphorylation sites (S262/S290/S324/S356E) reduced tau binding to HDAC6 (Fig. 4m). We next generated a constitutively active MARK2 construct (MARK2–T280E), which led to tau hyperphosphorylation at MARK2-target sites (S262/S356) and a striking dissociation of tau from HDAC6 (Fig. 4n). The abrogation of tau-HDAC6 binding was specific to MARK2, as the dissociation was not observed with GSK3β, a kinase that does not phosphorylate tau in the MTBR (Fig. 4n). MARK2-mediated tau phosphorylation not only led to the dissociation of the tau–HDAC6 complex, but also reduced tau and tubulin acetylation (Fig. 4o, p). Therefore, dissociation of tau from HDAC6 via tau kinase activation or by employing tau mutants, that abrogate HDAC6 binding (e.g., 4Δ) reduced the acetylation of tau (Fig. 4q and Supplementary Fig. 6b). This could again reflect the targeting of tau by other HDACs in a compensatory manner, since the MARK2-mediated tau deacetylation was TSA-sensitive (Fig. 4q). In contrast, we note that GSK3β, which promotes tau phosphorylation in regions flanking the MTBR, increased tau acetylation within the MTBR (Fig. 4o, p). Overall, our data suggest that tau kinases modulate the ability of HDAC6 to target and alter tau PTMs.

**Loss of HDAC6 accelerates tauopathy in mice.** In contrast to alleviating tauopathy progression, our results to this point suggest that HDAC6 depletion might enhance pathological tau and accelerate tauopathy in vivo. To directly test this hypothesis, we crossed P301S tau transgenic mice (PS19) to HDAC6 KO mice, thereby generating PS19 × HDAC6 KO mice (referred to hereafter as PHK mice). We chose the PS19 model since this line reliably recapitulates many hallmarks of tauopathy including robust tau pathology, cognitive dysfunction, and decreased survival[44]. We note that HDAC6 KO mice are breeding-competent, show no developmental or neurodegenerative phenotypes, live a normal lifespan, and show no tau pathology (Supplementary Fig. 7)[27,45]. The Kaplan–Meier survival curve showed the expected PS19 mortality rates similar to that previously reported (~50% survival by 12 months) (Fig. 5a, red line). The PHK mice showed accelerated mortality as early as 4 months old with significantly reduced survival at end stage (Fig. 5a, purple line). The reduced survival of PHK mice correlated with progressive loss in body weight (Fig. 5b).

To assess cognitive function, mice were evaluated in the Morris water maze for acquisition and strength of spatial learning. To exclude potential differences in motor function in tauopathy mice, we evaluated swim speed and observed no differences among all genotypes (Fig. 5c). During the hidden platform test, WT, HDAC6 KO, and PS19 mice showed significantly decreased latencies across the training period, while the PHK mice showed poor learning during the entire 9-day regimen (Fig. 5d). In a probe trial without the platform, only the WT and HDAC6 KO mice showed a preference for the correct quadrant (Fig. 5e, within genotype comparisons are denoted by #). Notably, a trend for quadrant preference was observed in the PS19 mice ($p = 0.0894$). In contrast, PHK mice failed to demonstrate any selectivity for the target quadrant and spent significantly more time than the other genotypes in the incorrect quadrant (Fig. 5e, purple). When swim path crosses were measured, the PHK mice also exhibited significantly fewer crosses over the target area, and more crosses over the incorrect location, in comparison to both WT and HDAC6 KO (Fig. 5f). Neither the PS19 or PHK mice had preference for crossing the target area, in contrast to WT and HDAC6 KO mice. Overall, the behavioral analysis suggests that PHK mice do not show acquisition of spatial learning, indicating that loss of HDAC6 exacerbates hippocampal-dependent cognitive decline in a mouse tauopathy model.

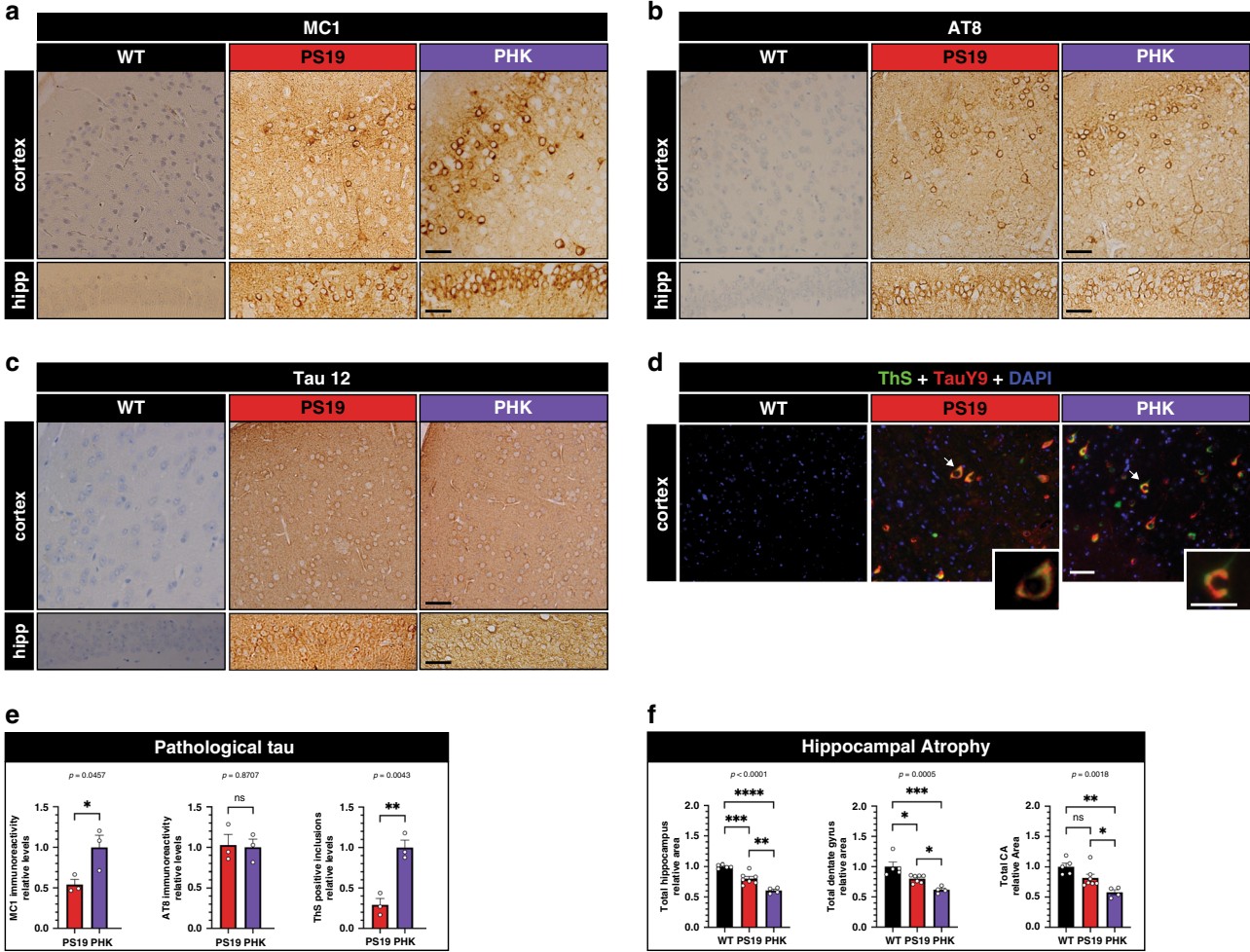

**Fig. 6 HDAC6 depletion accelerates tau pathology and neurodegeneration.** Representative IHC images of pathological tau detected with the MC1 (**a**), phosphorylated tau (AT8) (**b**), and total tau (Tau 12) (**c**) antibodies in the cortex (top panels) and hippocampus (bottom panels) of 12-month old WT, PS19, and PHK mice. Representative images are shown from $n = 3$ mice per genotype. Scale bar = 50 µm (**a–c**, scale bar is identical for both hippocampus and cortex). **d** ThS-positive tau pathology (green) is double-labeled with total tau (TAUY9, red) and DAPI (blue) to highlight cortical amyloid burden. The high magnification inset depicts a neuron with robust tau pathology. **e** Quantification of MC1 and AT8 immunoreactivity in the hippocampus and ThS-positive inclusions in the cortex ($n = 3$ mice per genotype). Error bars represent means ± SEM. Statistical analysis was performed using two-sided unpaired *t*-test and normalized to PHK. **f** Quantification of hippocampal area using total hippocampus, dentate gyrus, and CA brain area from 12-month-old mice ($n = 4$ mice per genotype). Measurements were performed in an unbiased manner using ImageJ software. Statistical tests: one-way ANOVAs were performed with Tukey's test for multiple comparisons among groups. Data shown represent means ± SEM normalized to WT. *$p < 0.05$; **$p < 0.01$; ***$p < 0.001$; ****$p < 0.0001$. Source data are provided as a Source Data file.

To evaluate the differential effects of HDAC6 depletion in the setting of Aβ plaques rather than tau pathology, we generated plaque-bearing mice lacking HDAC6 using the 5xFAD model of Aβ deposition (5xFAD × HDAC6 KO mice referred to hereafter as 5xFHK mice). Indeed, depletion of HDAC6 in the 5xFAD mice rescued the loss in body weight and spatial learning defects observed in 5xFAD mice (Supplementary Fig. 8), which is consistent with the previously reported cognitive improvements observed in APPPS1 mice upon HDAC6 depletion[27]. These results support the notion that HDAC6 loss has opposing effects depending on the nature of the pathology.

Given the cognitive defects in the tauopathy model, we next asked whether tau pathology was accelerated in PHK mice when compared to PS19 littermate controls. We evaluated pathological tau in the cortex and hippocampus by IHC analysis using the MC1 and AT8 antibodies, as well as Thioflavin S (ThS) fluorescence to evaluate the extent of mature tau amyloid structure[46,47] (Fig. 6a, d).

While total tau and phosphorylated tau immunoreactivity were comparable (Fig. 6b, c), PHK mice accumulated more MC1-immunoreactive tau in the soma and neurites of both the cortex and hippocampus (Fig. 6a, e). Consistent with the increased MC1 immunoreactivity, PHK mice also harbored more cortical ThS-positive tau inclusions when compared to PS19 mice (Fig. 6d, e and Supplementary Fig. 9a), indicating that loss of HDAC6 increases aggregate-prone tau species. Finally, we quantified hippocampal area as an indicator for neurodegeneration and observed a significant reduction in total area of the hippocampus, dentate granule layer, and CA regions in PHK mice compared to PS19 mice (Fig. 6f and Supplementary Fig. 9b).

Immunoblotting of mouse brain homogenates further showed that PS19 mice have minimal acetylated tau, while PHK mice lacking HDAC6 had elevated acetylated tau at multiple residues (K280 and K311) in parallel with increased tau phosphorylation within the MTBR (S262 and S356), but not epitopes outside the

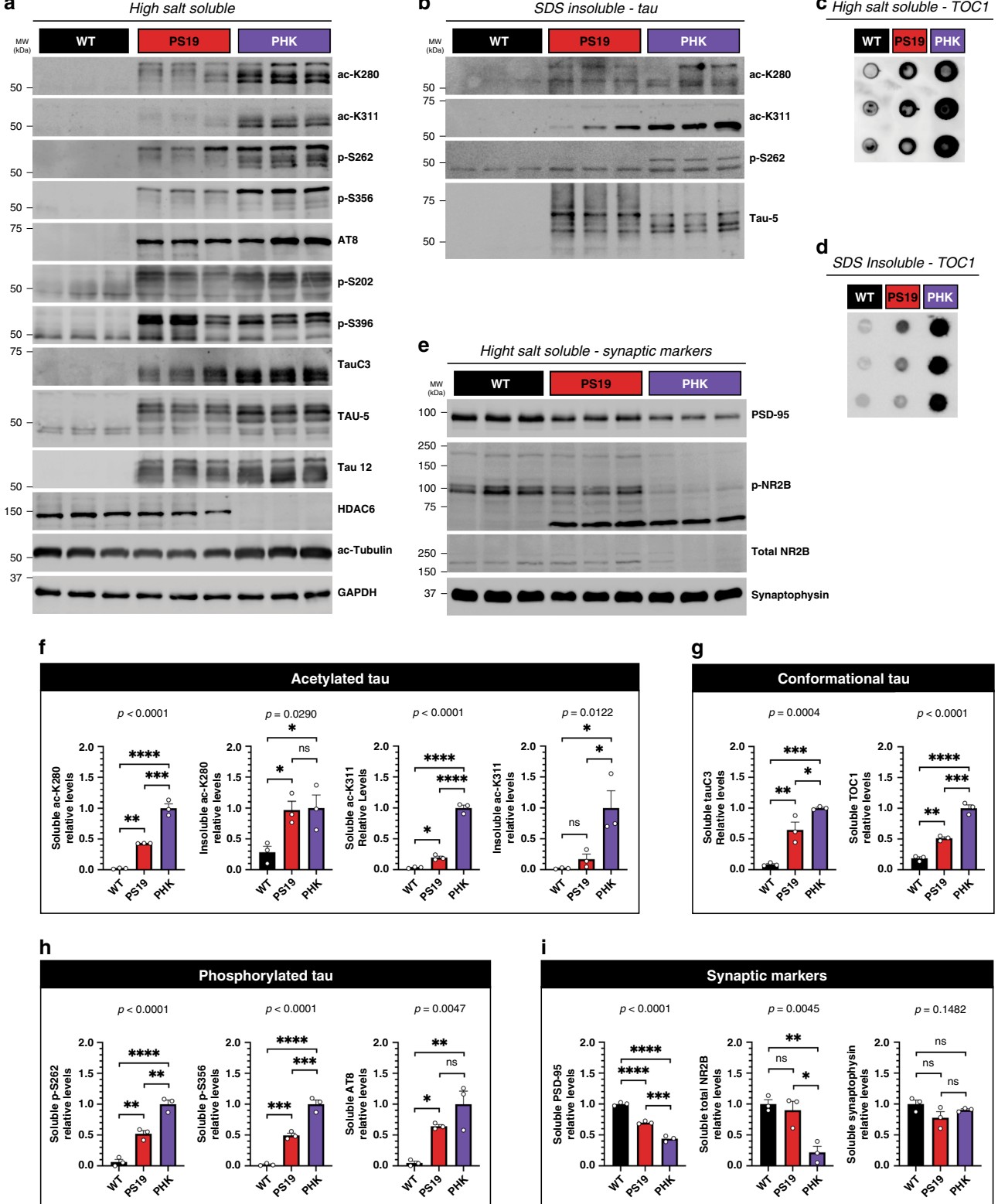

MTBR (AT8, S202, and S396) (Fig. 7a, b). In addition, pathological cleaved tau (detected by the TauC3 antibody) and tau oligomers (detected by the TOC1 oligomer-specific antibody) were also increased in PHK mice, supporting the notion that HDAC6 depletion generates a variety of pathological tau conformers (Fig. 7a–d). Lastly, consistent with tau-mediated neurotoxicity, the accelerated accumulation of tau pathology in PHK mice reduced postsynaptic (PSD-95), but not presynaptic (synaptophysin), markers (Fig. 7e). Quantification of the molecular changes in PHK mice is depicted in Fig. 7f–i, which strongly supports loss of HDAC6 as a mediator of tau pathogenesis, synaptotoxicity, and neurodegeneration.

**Fig. 7 HDAC6 depletion accelerates tau pathology and causes synaptic dysfunction. a** Immunoblotting of soluble high salt fractions from 12-month-old WT, PS19, and PHK cortex to evaluate the following: acetylated tau (ac-K280 and ac-K311), phosphorylated tau in the MTBR (p-S262 and p-S356), nonMTBR phosphorylated tau (AT8, p-S202, and p-S396), cleaved tau (TauC3), endogenous mouse tau (TAU-5, ~50 kDa), human tau (TAU-5 and Tau 12, ~65 kDa), HDAC6, and the HDAC6 substrate ac-tubulin. **b** Similar immunoblotting as in **a** was performed using insoluble SDS fractions. Acetylated tau (ac-K280 and ac-K311), phosphorylated tau in the MTBR (p-S262) and total tau (TAU-5) were detected in the insoluble fractions. **c, d** Dot blot analysis was used to detect oligomeric tau in soluble (**c**) and insoluble (**d**) fractions with the TOC1 antibody. **e** Synaptic markers including postsynaptic (PSD-95), NMDA receptors (p-NR2B and total NR2B), and presynaptic (Synaptophysin) were evaluated by immunoblotting. **f–i** Quantification of acetylated tau (**f**), conformational tau (**g**), phosphorylated tau (**h**), and synaptic markers (**i**) were determined by densitometry and values were normalized relative to PHK mice (**f–h**) or WT mice (**i**). Statistical tests: one-way ANOVAs were performed (*p*-values are listed above graphs) and Tukey's multiple comparison tests were used to compare among groups. Representative immunoblotting data are shown from $n = 3$ biologically independent samples (**a–e**). Error bars represent means ± SEM. *$p < 0.05$, **$p < 0.01$, ***$p < 0.001$, ****$p < 0.0001$, ns not significant. Source data are provided as a Source Data file.

## Discussion

In this study, we evaluated HDAC6 as a mediator of tau pathogenesis. Contrary to the notion that HDAC6 depletion alleviates neurodegeneration, our results suggest that HDAC6 acts in a protective capacity under conditions of tauopathy to suppress aberrant tau accumulation, and that sustained or chronic loss of HDAC6 results in accelerated tau pathology, cognitive dysfunction, and reduced survival. Overall, our data warrant a careful evaluation of whether targeting, depleting, or inactivating HDAC6 will provide therapeutic utility in either AD or other tauopathy patients. A more thorough analysis of the impact of HDAC depletion or inactivation in the brain is needed, including the cell-type specific implications of manipulating HDAC6, as well as clinical assessments at varying stages of disease progression.

Our prior study indicated that tau is a major target of HDAC6, as HDAC6 efficiently deacetylated tau in vitro and in cell-based assays[1]. We now demonstrate significantly increased acetylated tau in tauopathy mice lacking HDAC6, further validating HDAC6 as a bona fide tau deacetylase in vivo. Since soluble functional HDAC6 was depleted and co-aggregated with tau in AD brain, we speculate that gradual loss of HDAC6 function contributes to the accumulation of acetylated tau that we and others have commonly observed in AD brains, hence early changes in HDAC6 localization or activity may represent a pathological event that precedes tau pathogenesis. We note that HDAC3 and SIRT1 have also been implicated as deacetylases that regulate tau deacetylation (Fig. 1a)[13,48], suggesting potential synergy between HDAC6 and other deacetylases towards tau. Further analysis of mice lacking multiple deacetylases could address overlapping or compensatory roles for multiple HDACs in suppressing tau pathology.

While acute HDAC6 inhibition may be protective in the absence of tau pathology[18], our data in mouse and human brain suggest that chronic and/or sustained HDAC6 depletion may exacerbate any existing tauopathy leading to synaptic and cognitive defects. Furthermore, our data potentially reconcile a long-standing controversy as to whether loss of HDAC6 is beneficial or detrimental, as HDAC6 loss exacerbated disease progression in a tauopathy model (Fig. 5) but restored cognition in a model of Aβ plaque deposition (Supplementary Fig. 8). Thus, the specific nature of the pathology that emerges in the diseased brain may dictate whether HDAC6 is required. Loss or inhibition of HDAC6 may be neuroprotective under conditions in which tau pathology is absent, or at least below a particular threshold. Indeed, we previously showed that HDAC6 inhibition during an Aβ-induced neuroinflammatory challenge can suppress neuronal damage[40]. In the absence of robust tau pathology, tau is mostly bound to MTs[49] and not appreciably acetylated at most lysine residues[1,43]. Therefore, we suspect that HDAC6 inhibition under conditions lacking robust tau pathology might alter the acetylation status of non-tau HDAC6-dependent substrates (e.g., tubulin, cortactin,

Hsp90) and thus restore axonal transport and MT-dependent trafficking, for example in the presence of Aβ (as we observed in the 5xFAD model)[50]. Indeed, loss or inhibition of HDAC6 was reported to stabilize MTs and improve axonal transport in non-tauopathy syndromes including Charcot Marie Tooth (CMT) neuropathy[51,52], amyotrophic lateral sclerosis (ALS)[53], Rett syndrome[54], and Aβ plaque deposition[19].

In contrast, under conditions in which aberrant tau accumulates in the cytosol, either due to genetic mutations or other potential pathological triggers (e.g., MT dysfunction), HDAC6 activity may be required to bind tau via the aggregate-prone R2/R3 motifs and suppress aberrant tau PTMs that would otherwise generate toxic tau species. We propose that tau aggregation may in turn promote HDAC6 coaggregation and inactivation of its catalytic activity, which would further amplify this cycle and lead to robust acetylated tau accumulation, as we and others commonly observe in AD brains. In addition to tau, HDAC6 reversibly regulates components of the actin remodeling machinery (e.g., actin and cortactin)[55], a variety of MT-associated factors (e.g., tau, MAP2, and MAP4)[56], heat shock signaling (e.g., Hsp70 and Hsp90)[36,57], and more recently, stress granule (SG) formation by controlling liquid–liquid phase separation (LLPS)[58]. Therefore, HDAC6 inhibition could conceivably increase neuronal vulnerability by influencing tau processing as well as the stress response, mitophagy, autophagosome–lysosome fusion, and cytoskeletal remodeling. Future efforts are needed to more clearly determine the impact of HDAC6 inhibition on tau pathology, particularly at early versus late stages of tauopathy progression.

Using a panel of FTD-associated tau mutants, HDAC6 showed increased binding to P301L and S320F mutations. The exact properties of these particular mutants that enable more robust engagement with HDAC6 is not clear, however a recent study indicated synergistic aggregation and seeding of the P301L and S320F mutations[59–61]. Our data suggest that HDAC6 may preferentially target specific aggregate-prone conformations, or strains, via a chaperone bridge, as mutations that abrogate tau–chaperone binding also diminished tau–HDAC6 binding (Fig. 1l–n). The Hsc70/Hsp70 sensor appears to detect the R2/R3 motifs within the MTBR, thus recruiting and enabling HDAC6 to target particular tau species. How HDAC6 might also suppress tau phosphorylation at KXGS motifs is unclear, but this could involve a complex interplay between different tau remodeling complexes (e.g., tau targeted HDACs and kinases) and synergy amongst various tau PTMs. One intriguing possibility is that specific tau acetylation patterns prime phosphorylation within the MTBR, and therefore HDAC6-mediated deacetylation could effectively shift the equilibrium towards a less phosphorylated and presumably less toxic state. Indeed, a complex interplay among tau PTMs (e.g., acetylation and phosphorylation) within the MTBR was recently suggested[62]. Our data are consistent with specific priming events that establish a unique tau PTM profile, or signature, that results in tau-mediated toxicity.

Tau acetylation may facilitate the formation of toxic tau oligomers that lead to synaptic dysfunction. A K174Q acetylation-mimic mutant showed impaired tau turnover and induced cognitive deficits[3]. Similarly, K274/K281Q acetylation mimic transgenic mice showed altered levels of critical synaptic factors (e.g., KIBRA), altered postsynaptic remodeling, and impaired long-term potentiation leading to memory deficits[6]. Here, we observed that loss of HDAC6 similarly led to increased tau oligomers and correlated with reduced postsynaptic integrity (Fig. 7). Given the possibility that distinct prion-like tau strains could emerge under different clinical tauopathy settings, we propose that acetylated tau might contribute to, or perhaps even underlie, the formation of conformational tau variants in tauopathies. In support of this notion, we identified a potential 3R-tau specific strain (ac-K311 tau) that was prominent in AD and PiD, but not in CBD, PSP, and FTLD-tau cases harboring 4R-tau, even though the K311 residue is present in all six tau isoforms capable of forming tau inclusions (Fig. 3). Similarly, PS19 mice overexpressing human 4R-tau showed minimal ac-K311 immunoreactivity that was only significantly induced upon HDAC6 depletion (Fig. 7a). This indicates that 4R-tau, which has higher affinity binding to HDAC6 compared to 3R-tau (Fig. 1f–g), may preferentially suppress ac-K311 immunoreactivity. In this scenario, 3R-tauopathies would be marked by distinct acetylation profiles compared to 4R-tauopathies, providing a unique tau PTM signature that could explain the specific conformation and/or aggregation of distinct tau isoforms in clinically diverse neurodegenerative tauopathies (see Supplementary Fig. 10 model). This could also have important biomarker implications for distinguishing 3R from 4R tauopathies in patient biofluids.

In summary, aberrant tau acetylation could underlie a complex post-translational mechanism that operates in diverse human tauopathies. Our findings identify HDAC6 as critical neuroprotective factor that suppresses toxic tau accumulation. Future efforts to decipher the full tau PTM code linked to tau strain identity could extend our understanding of tau pathogenesis and lead to new avenues for personalized therapeutics in AD and related disorders.

## Methods

**Plasmids, cell culture, and transfection.** The human tau isoforms containing 0–2 N-terminal inserts and 3–4 MTBR repeats (2N4R) were cloned into pCDNA5/TO vector (Invitrogen). Site-directed mutagenesis was used to delete residues I277/I278/V308/V309 to generate tau 4Δ plasmids and also used to insert the indicated familial *MAPT* mutations. FLAG (FL)-tagged HDAC6 (WT, 1–840, ΔBUZ) and CBP expression plasmids (pcDNA3.1) were kindly provided by Dr. Tso-Pang Yao (Duke University). Site-directed mutagenesis was employed to generate H216A, H611A, and H216/611A mutants using the following primer pairs: HDAC6-216A-F, TCCTGGACATGCCGCCCAGCACAG; HDAC6-216A-R, GGCCTAAT-GATGGCCATG; HDAC6-H611A-F, CCCAGGACACGCCGCAGAGCAGGATG; HDAC6-H611A-R, GGACGCACCACAGCAGCA. The ΔSE14 deletion construct was generated using the following primers: dse14-F, CTAGCCTCGAGCACAGAC; dse14-R, GGTGACTTTCCCCATGCC. The tau 4Δ mutation was inserted into 2N4R tau or 2N4R-P301L tau using the following primers: d277/278-F, CCGGGAGGCGGGAAGGTGCAGAATAAGAAGCTGGATCTTAGC; d277/278-R, GCTAAGATCCAGCTTCTTATTCTGCACCTTCCCGCCTCCCGG; d308/309-F, CCCGGGAGGCGGCAGTGTGCAATACAAACCAGTTGACCTGAG; d308/309-R, CTCAGGTCAACTGGTTTGTATTGCA-CACTGCCGCCTCCCGGG. The Hsp27-GFP expression plasmid was purchased from Addgene (#17444). Hsc70-Myc, Hsp70-FL, and Hsp90-FL plasmids were kindly provided by Dr. Jonathan Schisler (University of North Carolina, Chapel Hill).

All described plasmids were transfected into HEK-293A cells using Fugene-6 (Promega) per the manufacturers protocols. HEK-293A cells (ThermoFisher, #R70507) are commercially available and maintained according to standard protocols. This cell line is a subclone of the standard HEK-293 line with a relatively flat morphology, is more slowly growing, and maintains ectopic plasmid expression at more physiological levels. Cells were harvested and analyzed by immunoblotting using the antibodies listed in Supplementary Table 2. The following deacetylase inhibitors (Sigma) were added overnight at the following final concentrations: 1 μM

trichostatin A (TSA, class I/II HDAC inhibitor), 5 μM tubastatin A (TBST, HDAC6 inhibitor), 5 mM nicotinamide (NCA, class III HDAC inhibitor).

**Mice and primary neuron cultures.** All protocols were carried out in accordance with the University of North Carolina (UNC) Institutional Animal Care and Use Committee (UNC IACUC protocol 19.017). Wild-type, HDAC6 KO, PS19, and 5xFAD strains (C57BL/6 background) aged 6–12 months old were used throughout this study for behavior analysis. Equal male/female cohorts were analyzed in all instances for behavior (only males were reported), histology, biochemistry, and survival data unless indicated otherwise. For embryonic neurons, 8 week-old female CD1 or C57BL/6 mice were used for timed pregnant breeding and cortical neurons were dissected from E16 embryos. Breeding conditions employ a 14-h light/10-h dark cycle with temperature range from 65 to 75 °F and 40–60% humidity. Primary cortical neurons were genotyped and cultured from E16 embryos using 8 week-old timed pregnant female CD1 mice (Charles River), PS19 (Jackson Laboratory, #24841), or tau KO (Jackson Laboratory, #7251) (for tau re-expression studies). Neurons were plated onto poly-D-lysine coated coverslips. Neurons were transfected with 0.5 μg of P301L Tau-GFP and 0.5 μg of HDAC6-FL plasmids, where indicated, using a calcium phosphate transfection kit in accordance with manufacturer instructions (Takara Bio). Coverslips were fixed with 4% paraformaldehyde (PFA) for 10 min, and cells were permeabilized by exposing coverslips to a 0.1% Triton X-100 in 1X PBS solution for 8 min. Blocking was performed for 1 h at room temperature in a solution of 2% milk in 1× TBS. Following blocking, coverslips were then incubated in primary antibody overnight at 4 °C followed by staining with an Alexa-conjugated secondary antibodies and counterstaining with DAPI. For HDAC6 inhibition in PS19 neurons, TBST was added overnight to neurons at 1–5 μM, where indicated, followed by harvest and lysis as explained in immunoblotting and biochemical methods.

**Immunoblotting and biochemical methods.** Biochemical analyses for preparation of lysates were performed as follows. Fractionation of cell lysates was performed by sequential extraction using buffers of increasing strength. Cells from 6-well culture dishes were scraped into 200 μl 1× RIPA buffer (50 mM Tris pH 8.0, 150 mM NaCl, 1% NP-40, 5 mM EDTA, 0.5% sodium deoxycholate, 0.1% SDS) containing 1 mM phenylmethylsulfonyl fluoride, a mixture of protease inhibitors (1 mg/ml pepstatin, leuptin, N-p-Tosyl-L-phenylalanine chloromethyl ketone, Nα-Tosyl-L-lysine chloromethyl ketone hydrochloride, trypsin inhibitor; Sigma), a mixture of phosphatase inhibitors (1 mM NaF, 1 mM sodium orthovanadate, and 1 mM beta-glycerophosphate; Sigma), and deacetylase inhibitors (2 μM TSA, 10 mM NCA). Samples were sonicated 20 times and centrifuged at $21,130 \times g$ for 30 min at 4 °C. Supernatant was collected as the soluble fraction, and the resultant insoluble pellets were extracted in urea buffer (7 M urea, 2 M Thiourea, 4% CHAPS, 30 mM Tris, pH 8.5), sonicated ten times and centrifuged at $21,130 \times g$ for 30 min at room temperature. Soluble and insoluble fractions were analyzed by western blotting using primary antibodies followed by secondary HRP-conjugated antibody detection (all secondary antibodies were used at 1:500 or 1:1000 dilution unless otherwise noted). Primary and secondary antibodies used are listed in Supplementary Table 2. Dr. Peter Davies (Feinstein Institute for Medical Research) provided the MC1 antibody and Dr. Tso-Pang Yao (Duke University) provided anti-HDAC6 and anti-HDAC10 antibodies. Images were captured using ImageQuant TL v8.1 software on an LAS 4000 imager (GE Healthcare). Densitometry measurements for immunoblots were performed on the ImageQuant TL Analysis Toolbox application, where immunoblot signals were background subtracted and then further normalized and used for quantification purposes as mentioned in figure legends. Relative levels represent densitometry values that have been background subtracted and then normalized to the appropriate sample group/genotype (set to 1.0) to generate a relative comparison among groups.

**Human brain fractionation.** Human control and AD Braak V–VI cases were provided by the Center for Neurodegenerative Disease Research (CNDR) at the University of Pennsylvania. Consent for autopsy was obtained from legal representatives for all subjects in accordance with local institutional review board (IRB) requirements. Demographics of all cases used for immunoblotting, IHC and double-labeling IF analysis are described in Supplementary Table 1. Isolated gray matter from frontal cortex was homogenized in 3 vol/g of cold High Salt RAB buffer (0.75 M NaCl, 100 mM Tris, 1 mM EGTA, 0.5 mM MgSO4, 0.02 M NaF, 2 mM DTT, pH 7.4). All buffers were supplemented with protease inhibitor cocktail, phosphatase inhibitors, and deacetylase inhibitors. Homogenates were incubated at 4 °C for 20 min to depolymerize MTs, then centrifuged at $100,000 \times g$ for 30 min at 4 °C. Pellets were rehomogenized and centrifuged in 3 vol/g of cold High Salt RAB buffer. Resultant pellets were homogenized in 5 vol/g of cold RIPA buffer (50 mM Tris pH 8.0, 150 mM NaCl, 1% NP-40, 5 mM EDTA, 0.5% sodium deoxycholate, 0.1% SDS) and centrifuged at $100,000 \times g$ for 30 min at 4 °C. Myelin flotation was performed on pellets re-extracted in RIPA buffer supplemented with 20% sucrose. Finally, resultant insoluble pellets were extracted in 1 vol/g tissue in urea buffer (7 M urea, 2 M Thiourea, 4% CHAPS, 30 mM Tris, pH 8.5). Soluble and insoluble fractions were analyzed by SDS-PAGE electrophoresis and immunoblotting using the antibodies described in Supplementary Table 2.

**Coimmunoprecipitation assay**. For coimmunoprecipitation studies, HEK-293A cells were lysed in NP-40 lysis buffer referred to as NETN buffer (50 mM Tris-Cl, pH 7.4, 150 mM NaCl, 1 mM EDTA, 1% NP-40, 1 mM sodium orthovanadate, 1 mM sodium fluoride, 1 mM PMSF, 0.2 mM leupeptin, protease inhibitor cocktail, 2 μM TSA, 10 mM NCA). Soluble supernatants (1 mg total protein) were cleared by centrifugation, incubated overnight with protein A/G beads (Santa Cruz) complexed to the indicated antibodies described in the figure legends, and subsequently analyzed by SDS-PAGE and immunoblotting. Quantification of Co-IP assays were as described in the immunoblotting and biochemical methods. Values represent a ratio of immunoprecipitated tau to the total tau input, background subtracted, and normalized to WT.

**Antibody generation**. Polyclonal anti-acetylated tau antibodies (ac-K311) were generated using the acetylated tau peptide [307]QIVYK[Ac]PVDLSKVTSC[320] containing acetylated K311 to immunize rabbits (Genscript, Piscataway, NJ). Double affinity purification was performed using native and acetylated peptides. Site specificity of ac-K311 was confirmed first by peptide ELISA assays using acetylated and unmodified peptides, and then by immunoblotting analysis of purified WT and K311R mutant tau proteins. Highly specific polyclonal preparations from four different rabbits were used to confirm our findings in all of the described human studies. To validate antibody epitope reactivity, polyclonal #5089 affinity for acetylated tau was examined and showed only minor cross-reactivity with the K311R nonacetylated mutant (Supplementary Fig. 4), and therefore was used most extensively throughout this study (denoted as ac-K311).

**Mouse perfusions, fixation, and tissue processing**. This study was performed in strict compliance with animal protocols approved by the Institutional Animal Care and Use Committees (IACUC) of the University of North Carolina at Chapel Hill (UNC) under an approved protocol (#19.017). Mice were anesthetized with isoflurane and all efforts were made to minimize pain suffering. Mice were transcardially perfused with 15 ml of phosphate-buffered saline (PBS), and brains were removed, followed by immersion fixation for 48 h with 10% formalin in 1× PBS and paraffin embedding, where indicated. Paraffin blocks were sectioned at 5 μm thickness for IF and IHC analyses. Mouse brain tissue harvested for the purposes of perfusion-fixation, brain tissue fractionation, or dot blotting was performed on 12-month-old mice using littermates as controls.

**Immunofluorescence (IF) microscopy**. Double-labeling IF analyses were performed using Alexa Fluor 488- and 594-conjugated secondary antibodies (Molecular Probes, Eugene, OR). The sections were then treated to remove autofluorescence with Sudan Black solution for 5 min, and secured on coverslips with Vectashield mounting medium (Vector Laboratories). Digital images were obtained using an Olympus BX 51 microscope (Tokyo, Japan) equipped with bright-field and fluorescence light sources using a ProgRes C14 digital camera (Jenoptik AG, Jena, Germany) and Adobe Photoshop, version 9.0 (Adobe Systems, San Jose, CA) or digital camera DP71 (Olympus) and DP manager (Olympus).

For Thioflavin-S (ThS) staining, sections were deparaffinized and hydrated to dH$_2$O as mentioned above, followed by antigen retrieval and were rinsed in PBS, immersed in 0.05% KMnO$_4$/PBS for 4 min, and destained in 0.2% K$_2$S$_2$O$_5$/0.2% oxalic acid/PBS, immersed in 0.0125% ThS/40% EtOH/60% PBS and differentiated in 50% EtOH/50%PBS for 10–15 min. After differentiation, sections were treated for autofluorescence with Sudan Black solution for 30 s and quickly rinsed with 70% EtOH followed by wash with dH$_2$O, incubation with DAPI, and secured on coverslips. IF analyses were performed using Alexa Fluor 488-conjugated and 594-conjugated secondary antibodies (Molecular Probes) and imaged with Olympus IX83 inverted microscope. To confirm co-localization experiments, all relevant images were separately and independently imaged on a Zeiss LSM 780 confocal microscope. In some instances, background noise was removed from the confocal z-stacks using the denoise function in Nikon NIS Elements software. Maximum intensity projections of the denoised images displayed in figures were generated in FIJI/ImageJ. All quantitative fluorescence was independently validated with a minimum of $n = 3$ independent biological replicates.

**Immunohistochemistry (IHC) analysis**. For human studies, fixed, paraffin-embedded tissue blocks from the midfrontal cortex were obtained from the Center for Neurodegenerative Disease Research (CNDR) Brain Bank at the University of Pennsylvania. The demographics of all cases are listed in Supplementary Table 1. Consent for autopsy was obtained from legal representatives for all subjects in accordance with local institutional review board requirements. Briefly, fresh tissues from each brain were fixed with 70% ethanol in 150 mM NaCl, infiltrated with paraffin, and cut into 6-μm serial sections. IHC was performed using the avidin-biotin complex (ABC) detection system (Vector Laboratories, Burlingame, CA) and 3,3′-diaminobenzidine. Briefly, sections were deparaffinized and rehydrated, antigen retrieval was done by incubating section in 88% formic acid for 5 min, endogenous peroxidases were quenched with 5% H$_2$O$_2$ in methanol for 30 min, and sections were blocked in 0.1 mol/L Tris with 2% fetal bovine serum for 5 min. Primary antibodies were incubated overnight at 4 °C. After washing, sections were sequentially incubated with biotinylated secondary antibodies for 1 h and ABC for 1 h. Bound antibody complexes were visualized by incubating sections in a solution containing 100 mM TrisHCl, pH 7.6, 0.1% Triton X-100, 1.4 mM diaminobenzidine, 10 mM imidazole, and 8.8 mM H$_2$O$_2$. Sections were then lightly counterstained with hematoxylin, dehydrated, and mounted on coverslips. We note that IHC analysis of human tissue using acetylated tau antibodies requires ethanol fixation and antigen retrieval (formic acid) for maximal detection sensitivity of tau pathology. Double-labeling IF experiments on human tissue were performed with antigen retrieval of 88% formic acid for 5 min followed by incubation with primary antibodies overnight at 4 °C. Secondary Alexa Fluor 488 and 594 species-specific conjugated secondary antibodies (Molecular Probes) were then incubated overnight at 4 °C. Slides were treated for autofluorescence using a 0.3% Sudan Black solution and secured by coverslips with Vectashield-DAPI mounting medium (Vector Laboratories).

For mouse brain IHC analysis, we used the avidin–biotin complex (ABC) detection system (Vector Laboratories) and 3,3′-diaminobenzidine. Briefly, sections were deparaffinized and rehydrated immersed in 1× PBS pH 7.4 for 1 h. Antigen retrieval was done by boiling sections in citrate buffer pH 6.0, endogenous peroxidases were quenched with 5% H$_2$O$_2$ in water for 30 min and sections were blocked in 0.1 M Tris with 2% fetal bovine serum for 5 min. Primary antibodies were incubated overnight at 4 °C. After washing, sections were sequentially incubated with biotinylated secondary antibodies for 1 h and ABC complex for 1 h. Bound antibody complexes were visualized by incubating sections in DAB substrate (Sigma, D3939) a solution containing 100 mM TrisHCl, pH 7.6, 0.1% Triton X-100, 1.4 mM diaminobenzidine, 10 mM imidazole and 8.8 mM H$_2$O$_2$. Sections were then lightly counterstained with hematoxylin, dehydrated, secured on coverslips, and imaged using an Olympus BX41 microscope. The entire cortex and hippocampus (20×) for atrophy measurements were imaged using a Nikon EclipseTi2 inverted microscope. To assess the extent of tau pathology by IHC in mouse brain sections, quantification of MC1, AT8, and ThS-positive neurons was performed by counting tau-positive neurons in the hippocampus of 12-month-old mice from $n \geq 3$ mice per genotype.

**Mouse brain fractionation**. Fractionation of mouse brain was performed by sequential extraction using buffers of increasing strength. Mice brain tissue was homogenized in four volumes per gram of high-salt buffer (10 mM Tris base, 500 mM NaCl and 2 mM EDTA) supplemented with deacetylase, phosphatase and protease inhibitors (2 μM TSA, 10 mM NCA, 1 mM NaF; 1 mM Na$_3$VO$_4$; 1 mM PMSF; and protease inhibitor cocktail) and centrifuged at $21,130 \times g$ for 45 min to generate high-salt fractions. Resulting pellets were re-extracted in four volumes per gram of RIPA buffer (50 mM Tris pH 8.0, 150 mM NaCl, 1% NP-40, 5 mM EDTA, 0.5% sodium deoxycholate, 0.1% SDS). Myelin floatation was performed on pellets re-extracted in RIPA buffer supplemented with 20% sucrose. Finally, resultant insoluble pellets were extracted in 1 vol/g of tissue in SDS buffer (1% SDS in 50 mM Tris, 150 mM Nacl, pH 7.6). Lysates from soluble and insoluble fractions were analyzed by SDS-PAGE electrophoresis and immunoblotting using the indicated antibodies from Supplementary Table 2.

**Dot blotting**. Mouse brain homogenates were prepared as described above. Nitrocellulose membranes were soaked in water for 10 min before 10 μl of samples were spotted directly on to nitrocellulose membrane using the Bio-Dot apparatus (BioRad), and incubated at room temperature for 1 h, blocked in 2% milk in 1× TBS for 1 h. Membranes were incubated in the primary antibody TOC1 (1:500)[63] overnight at 4 °C followed by incubation with IgM-HRP-conjugated secondary antibody for 1 h at room temperature and imaged.

**Immunoprecipitation and mass spectrometry analysis**. Mass spectrometry analysis was performed at the University of Pennsylvania proteomics core facility, which identified sites of tau acetylation and phosphorylation. Immunoprecipitation followed by mass spectrometry was performed as follows. AD cortical lysates were prepared as described above and soluble high salt fractions containing all six tau isoforms were used as starting material. Total tau proteins were immunoprecipitated with pooled monoclonal T14 and T46 antibodies and separated by SDS-PAGE followed by gel excision and nanoLC/nanospray/MS/MS at the University of Pennsylvania proteomics core facility using LTQ XL* Linear Ion Trap Mass Spectrometer (Thermo Scientific). Data was acquired using Xcalibur software (Thermo Scientific) and analyzed by Mascot, Scaffold, and PEAKS 6.0 (Bioinformatics Solutions Inc.) software packages. Significantly modified peptide cutoffs showed $p$-values < 0.05 and peptide scores >20. The modified peptides isolated from three independent AD cases with significant scores were pooled and displayed in Table 1.

**Mouse behavior**. PS19 transgenic mice harbor a P301S mutant 1N4R-tau transgene driven by the mouse prion promoter (PrP). Heterozygous PS19 mice on a pure C57BL/6J background were acquired from Dr. Virginia Lee (University of Pennsylvania) and crossed to HDAC6 KO mice acquired from Dr. Tso-Pang Yao (Duke University)[64], to generate the desired PS19-HDAC6 KO mice on a congenic C57BL/6J background (PHK mice). The PS19 model does not show any sex-specific differences in neurodegeneration or tau pathology, and therefore we used male PS19 mice for all behavior analysis. 5xFAD mice harbor mutations in familial AD genes (APP and PSEN1) and produce abundant widespread plaques that lead

to cognitive deficits[65]. 5xFAD mice are commercially available from Jackson Laboratories (#34848) and were similarly crossed to HDAC6 KO mice to generate the desired 5xFAD–HDAC6 KO mice (5xFHK). For consistency, we similarly analyzed male 5xFAD mice for all behavior analysis, but verified our results with mixed cohorts including both male and female 5xFAD mice that yielded similar results to those reported in this study. Both PS19 and 5xFAD were bred and analyzed as heterozygotes and compared with littermates lacking one or both HDAC6 alleles (we note that HDAC6 is an X-linked gene).

The Morris water maze was used to evaluate spatial learning and memory. Testing was conducted by personnel that were blinded to genotype. The maze, a large circular pool (diameter = 122 cm) partially filled with water (45 cm deep, 24–26 °C), was located in a room with numerous visual cues. The procedure involved an initial visible platform test for vision and swimming ability, followed by a hidden platform task for acquisition of spatial learning. For the visible platform test, each mouse was given four trials per day, across 3 days, to swim to an escape platform cued by a patterned cylinder extending above the surface of the water. For each trial, the mouse was placed in the pool at 1 of 4 possible locations (randomly ordered), and then given 60 s to find the visible platform. If the mouse found the platform, the trial ended, and the animal was allowed to remain 10 s on the platform before the next trial began. If the platform was not found, the mouse was placed on the platform for 10 s, and then given the next trial. Measures were taken of latency to find the platform and swimming speed via an automated tracking system (Noldus Ethovision). Following the visible platform task, mice were tested for their ability to find a submerged, hidden escape platform (diameter = 12 cm). Each mouse was given four trials per day, with 1 min per trial, to swim to the hidden platform. After 9 days of training, mice were given a 1 min probe trial in the pool with the platform removed. Selective quadrant search was evaluated by measuring percent total time spent in the target quadrant (where the platform had been located during training) versus the opposite quadrant, and number of swim-path crosses over the target location versus the corresponding area in the opposite quadrant. Behavior analysis was performed in 7–9-month-old mice from PS19 and 5×FAD cohorts. In instances where mice were sacrificed after behavior analysis for perfusion fixation and tissue harvest 12-month-old mice were used.

**Recombinant tau protein purification**. Purification for the following K18 fragment and full length T40 recombinant tau proteins were performed, as described below. Protein expression, extraction, and purification were performed using chromatography methodology to purify heat stable tau proteins. Tau-K18 and tau-T40 plasmids were cloned into the pRK172 bacterial expression vector for inducible protein expression. Protein was expressed in BL21 (DE3) RIL E. coli cells. Bacteria were grown in lysogeny broth, ampicillin was added, and when an OD of 1.0 was reached, protein expression was induced with isopropyl-β-D-thiogalactopyranoside at a final concentration of 1.0 mM. After continued growth for 2 h, bacterial cultures were then centrifuged at $2340 \times g$ for 15 min, and pellets were immediately frozen at −80 °C. Pellets were resuspended in a high salt RAB buffer, pH 7.0 (0.1 M MES, 1 mM EGTA, 0.5 mM MgSO$_4$, 750 mM NaCl, 20 mM NaF, 0.1 mM PMSF, 0.1% protease inhibitor cocktail). This resuspension was homogenized, boiled, and centrifuged at $100,000 \times g$ for 30 min. The resulting supernatant, with addition of 0.1 mM PMSF and 0.1% protease inhibitor cocktail, was dialyzed against FPLC buffer, pH 6.5 [20 mM piperazine-N,N′-bis(ethanesulfonic acid), 10 mM NaCl, 1 mM EGTA, 1 mM MgSO$_4$, 0.1 mM PMSF and 2 mM DTT]. After overnight dialysis, the contents were passed through a HiTrap sulfopropyl sepharose high performance cation exchange column (GE) attached to an ÄKTA Pure chromatography system equilibrated in FLPC buffer. Fractions were eluted over a 0–0.4 M NaCl gradient. Portions of the fractions were separated by SDS-PAGE and stained with Coomassie Blue R-250 (Fisher BioReagents). Fractions containing tau protein were subsequently pooled. The FPLC buffer present in pooled tau protein fractions was exchanged for 100 mM sodium acetate (pH 7.0), and protein was concentrated using Amicon Ultra centrifugal filter devices (Millipore Corporation, Billerica, MA). Resultant protein concentration was determined using bicinchoninic acid assay (Thermo Scientific Pierce).

**HDAC in vitro activity assays**. In vitro recombinant deacetylase activity was measured using the HDAC1 (BML-AK511), HDAC3 (BML-AK531), or HDAC6 (BML-AK516) FLUOR DE LYS fluorometric activity assay kits, per the manufacturer's detailed protocol and instructions (Enzo Life Sciences). We assessed the inhibition of deacetylase activity with the following recombinant full-length 2N4R tau proteins: WT, P301L, L315R, S320F, and ΔR1–4 lacking the MTBR. All full-length tau proteins were added to a final concentration of 4.36 μM, while HDACs were pre-loaded at the following concentrations: HDAC1, 0.18 μM; HDAC3, 4.08 μM; HDAC6, 74.63 nM per the manufacturer's protocol. Samples lacking tau but containing HDACs and substrate represent negative controls for HDAC inhibition (100% HDAC activity). Trichostatin A (TSA) was used as a positive control for inhibition of HDAC activity and was employed at a final concentration of 10 nM. Deacetylase reactions were carried out for 1 h at 30 °C, followed by the addition of developer for 45 min. Each individual HDAC assay was plotted as normalized HDAC activity relative to its respective TSA-treatment, which was established as a relative baseline representing maximal inhibition of HDAC activity. Fluorescence measurements were acquired using the FLUOstar Omega microplate reader (BMG LABTECH, Germany).

**Tau fibrillization assay**. The extent of heparin induced aggregation was assessed for the following recombinant K18 tau proteins: WT, P301L, K311Q, and K311R. For each K18 tau protein, a master mix fibrillization reaction was generated and then aliquoted into 50 μl for each time point. Fibrillization reactions were generated in the following manner: 10 μM K18 tau proteins were prepared in 100 mM sodium acetate (pH 7.0) buffer followed by 2 mM DTT and then 10 μM heparin, upon incubation at 37 °C. At each time point, samples were either immediately centrifuged at $21,130 \times g$ at 4 °C for the sedimentation assay, frozen at −80 °C for ThT fluorescence, or processed for DLS.

**Sedimentation assay**. At specified time points, fibrilization reactions used for the sedimentation assay were immediately removed from 37 °C incubation and centrifuged at $21,130 \times g$ for 30 min at 4 °C to separate the soluble supernatant and insoluble pellet fractions. Samples for the supernatant fractions were combined with 6× loading buffer and 100 mM DTT, while the pellets were resuspended with 1× loading buffer and 100 mM DTT. Equal amounts of protein for each reaction and fraction, were boiled for 10 min at 98 °C and separated on 15% handcast SDS-PAGE gels. Gels were subsequently stained with Coomassie Brilliant Blue R-250 (Fisher BioReagents) for 30 min and destained overnight for imaging.

**Thioflavin T (ThT) assay**. At specified time points, fibrilization reactions used for the ThT assay were immediately removed from 37 °C incubation and frozen at −80 °C. For the stock ThT solution, ThT was freshly prepared at 1 mM dissolved in 100 mM sodium acetate buffer, pH 7.0 and filtered to a final working solution of 10 μM ThT. In each well, 15 μl of the fibrilization reaction (or buffer) was combined with 140 μl of 10 μM ThT for triplicate measurements per fibrilization reaction and time point. Samples were measured at 430 nm excitation and 500 nm emission using the FLUOstar Omega microplate reader (BMG Labtech, Germany).

**Dynamic light scattering (DLS)**. Measurements were performed at 37 °C using a DynaPro Dynamic Light Scattering Plate Reader (Wyatt Technology). Freshly prepared fibrilization reactions were centrifuged at $18,390 \times g$ for 5 min to remove any debris, prior to loading 100 μl of sample. For each timepoint after induction of tau fibrilization reactions, 10 acquisitions were taken and analyzed using DYNAMICS 7.1.7.16 Software. DLS was used to monitor the presence of oligomeric tau species, while ThT and aggregation assays are commonly used to measure fibrillar tau aggregates that contain β-structure.

**Statistics and reproducibility**. For animal behavior studies, results were first analyzed using repeated measures analysis of variance (ANOVA). Fisher's protected least-significant difference (PLSD) tests were used for comparing group means. Within-genotype repeated measures analyses were used to determine learning across days of training in the hidden platform procedure, and quadrant selectivity in the probe test. For all comparisons, significance was set at $p < 0.05$. Survival analysis was calculated using a log-rank (Mantel–Cox) test to assess significance among genotypes (WT, $n = 22$; HDAC6 KO, $n = 42$; PS19, $n = 27$; PHK, $n = 37$). For the behavioral studies involving tauopathy mice, 7–8 month old PS19 male mice were analyzed (WT, $n = 11$; HDAC6 KO, $n = 16$; PS19, $n = 15$; PHK, $n = 13$), while 7–9 month old 5xFAD male mice were analyzed (WT, $n = 7$; HDAC6 KO, $n = 7$; 5xFAD, $n = 8$; 5×FHK, $n = 10$). Hippocampal atrophy measurements were determined by unbiased tracing of the dentate granule layer, CA regions, or the entire hippocampus using image-J software (WT, $n = 5$; PS19, $n = 7$; PHK, $n = 4$). All cross-genotype comparisons involving three or more genotypes were analyzed using one-way ANOVAs with Tukey's posthoc analysis for comparisons between the means of each group.

Significant differences from co-IP experiments, cell culture immunoblotting experiments, and HDAC activity assays were determined by two-tailed unpaired t-test from $n = 3$ independent biological replicates. Significant differences from immunoblotting of mouse brain lysates were determined by one-way ANOVA with Tukey's multiple comparison or t-test with Welch's correction from a minimum of $n = 4$ mice per age group per genotype. Significant differences in protein expression by immunoblotting of human brain lysate was determined from $n = 5$ AD brains using two-sided unpaired t-test with Welch's correction. The number of tauopathy cases analyzed by immunoblotting, IHC, and IF are described in Supplementary Table 1.

**Reporting summary**. Further information on research design is available in the Nature Research Reporting Summary linked to this article.

## Data availability
The authors declare that all data supporting the findings of this study are available within the paper and its supplementary information files. The complete source data in this manuscript are provided as a Source Data file. Source data are provided with this paper.

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

## Acknowledgements

Support for this work was provided by the National Institutes of Health (NIH) grants R21AG058080 (T.J.C.), K23NS088341 (D.J.I.), National Center for Advancing Translational Sciences (NCATS) grant UL1TR001111 (T.J.C.), Alzheimer's Association grant NIRG-14-321219 (T.J.C.), Association for Frontotemporal Degeneration (AFTD) pilot grant (T.J.C.), American Federation for Aging Research (AFAR) grant RAG15247 (T.J.C.), CurePSP grant 656-2018-06 (T.J.C. and D.J.I.), and NSF Graduate Research Fellowship Program (GRFP) grant DGE-1650116 (H.T.). The Mouse Behavioral Phenotyping Core and the Neuroscience Microscopy Core Facility (RRID:SCR_019060) are supported by the NIH-NICHD Intellectual and Developmental Disabilities Research Center Grant U54 HD079124. The Neuroscience Microscopy Core Facility is also supported by funding from the NIH-NINDS Neuroscience Center Grant P30 NS045892. M.S.I. is supported by an Imaging Scientist grant from the Chan Zuckerberg Initiative. This work was also supported by the National Cancer Institute of the NIH award P30CA016086. The content is solely the responsibility of the authors and does not necessarily represent the official views of the NIH. We would like to thank Dr. Peter Davies (Feinstein Institute for Medical Research) for providing the MC1 antibody, Dr. Nicholas Kanaan (Michigan State University) for providing the TOC1 antibody, Dr. Jonathan Schisler for providing Hsp70 and Hsp90 plasmids, and Dr. Tso-Pang Yao (Duke University) for providing HDAC6 KO mice, HDAC6 expression plasmids, and anti-mouse HDAC6 and HDAC10 antibodies. We would also like to thank the UNC Animal Histopathology core, the UNC Neuroscience Microscopy core, and Dr. Xu Tian for providing excellent technical support.

## Author contributions

H.T. led most aspects of the experimental study and performed the molecular, cell and neuron based experiments, and biochemical experiments including mass spectrometry, human tissue analysis, protein purification, and mouse immunoblotting/data analysis. D.A. performed mouse histology experiments and assisted with immunoblotting. J.H.T. performed HDAC6 assays, human brain I.F., primary neuron culture, and assisted with protein purification. Y.C. assisted with plasmid preparation, mouse handling, breeding, genotyping, perfusions, and colony maintenance. A.A. and Z.T. provided technical assistance with immunoblotting, mass spectrometry analysis, and atrophy measurements. R.L. and C.P. performed human tissue I.H.C. and double labeled I.F. experiments under supervision of D.J.I. Human histology samples were reviewed and rated by D.J.I. V.M.Y.L. and J.Q.T. provided human tissues and biochemically isolated P.H.F.-tau extracts from human cases. S.S.M. and N.V.R. performed all mouse behavior and analysis. A.T. assisted H.T. with DLS assays. M.S.I. and H.T. performed confocal imaging to confirm co-localization. H.T. and T.J.C. co-wrote the manuscript. This study was directed and supervised by T.J.C.

## Competing interests

The authors declare no competing interests.
