## [Peer Review File · Nature Communications]

Reviewers' comments:

Reviewer #1 (Remarks to the Author):

Overall an interesting study which I believe will be of interest to the field. The manuscript is written well and is nicely presented. For publication, however, several claims need to be strengthened with additional experiments and/or appropriate quantification.

Comments:

1. In figure 1 the authors mention 'binding affinity' several times yet only co-immunoprecipitation has been conducted (page 5 line 12; page 6 line 1; page 6 line 11, page 6 line 17; page 6 line 23; page 7 line 4; page 7 line 15; page 7 line 22; page 7 line 23). Co-IP cannot determine the affinity between two proteins and if the authors wish to make claims regarding affinity or whether the binding between two proteins is reduced or increased, then surface plasmon resonance needs to be conducted (or the Blitz system utilized).
2. Were the images in Figure 2 captured using a confocal microscope with z-stacks? If only a fluorescence microscope was used then the term co-localisation should not be used
3. Page 13 line 3: the authors mention that S202 and T205 p-epitopes don't increase yet in Figure 4d there is a significant increase in these levels
4. Page 13 line 4: the authors mention that there is reduced tau phosphorylation at MARK2 sites S262 and S356 but S396 is also reduced in figure 4d
5. For Figure 4, what were acetylated and p-tau levels normalized to? If acetylation of tau increases the half-life of tau then post-translationally modified tau levels should be normalized to total tau rather than a housekeeping protein
6. Figure 6a-d: quantification needs to be conducted to support claims
7. Figure 6e: can you please include the images for this data
8. Figure 7a-b: please see comment 6 above
9. Page 17 line 20: " We now demonstrate significantly increased acetylated tau in tauopathy mice lacking HDAC6...." Please refer to comment 6
10. Page 19 line 18, "... Increased affinity to P301L and S320F mutations." Needs to be verified with additional experiments.
11. Unpaired t-tests were used frequently for statistical analysis of more than two groups- please repeat statistical analysis of these groups with a one-way ANOVA

Reviewer #2 (Remarks to the Author):

In a previous study, the authors of this manuscript have identified HDAC6 as a tau deacetylase and have shown that acetylation of tau inhibits its function and promotes pathological tau aggregation (Cohen et al. 2011, Nat. Comm.). In this study they investigated the role of HDAC6 in tauopathy progression. The first part deals with the molecular interplay between tau and HDAC6. Here, the authors identified a complex of HDAC6 and HSP70 targeting tau via the microtubule binding region in vitro and claimed that HDAC6 accumulates in pathological tau lesions in mice and

human AD brain tissue. Using a new ac-K311 antibody the authors then show that hyperacetylation of tau is preferentially occurring in 3R-tauopathies compared to 4R-tauopathies. Finally, they show that there is an interplay between HDAC6-mediated deacetylation and the phosphorylation status of tau. In the second part of the manuscript, the authors have used mouse models to investigate the effect of HDAC6-KO on tauopathies. Using a newly created mouse line called PHK (PS19 & HDAC6-KO), they observed a decline in cognitive functions as well as accelerated tau pathology and neurodegeneration in HDAC6 deficient mice (PHK) when compared to the tauopathy model (PS19).

With this study the authors tackle an ongoing and controversial question, namely "is HDAC6 activity protective or detrimental in neurodegenerative diseases?". The assumption that depleting or inactivating HDAC6 activity might have detrimental effects is not new to the field (see review: Simoes-Pires et al. 2013, Mol. Neurodegen.). The authors of this study, however, add new knowledge about underlying mechanisms of the interplay between HDAC6 and tau in tauopathy models. The manuscript is generally well written, the choice of models and methods is appropriate and the claims of the study are novel and nicely build upon previous findings of the authors. However, the role of acetylated tau, including the interplay of tau and HDAC6, has been characterized in various studies before (see recent review: Naseri et al. 2019, Neurosc. Letters, 10.1016/j.neulet.2019.04.022). Hence, the authors should point out the novelty of their findings in regard to previous findings in more detail.

In summary, there are several weaknesses as pointed out below that need to be addressed before this manuscript can be recommended for publication in Nature Communications.

Major comments:

The authors call their manuscript "An HDAC6-dependent surveillance mechanism suppresses tau-mediated neurodegeneration and cognitive decline". However, this title is misleading because what this study shows is that transgenic mice lacking HDAC6 showed accelerated tau pathology, including neurodegeneration and cognitive decline. Therefore, the title should be adapted.

A major claim of this study is that 3R tau is selectively acetylated. However, the data presented here is not strong enough to support this notion (see comments below), especially because the authors mention that in in vitro studies 3R and 4R tau isoforms are identically acetylated by CBP. Fig. 3c: In order to evaluate aberrant acetylation of tau in 3R and 4R tauopathies the authors should add a figure showing the level of acK311 in normal healthy brain.

Fig. 3d: The authors claim to show colocalization of Tau and ac-K311 in 3R tauopathies (AD and PiD), however also 4R-tau in AD colocalized with ac-K311. The authors should mention and discuss this. In the third panel (PiD – RD3) both, tau and acK311 signals are highly overexposed. The authors should use different images to show that these staining are not artifacts. Also, the authors should describe how they have tested the specificity of anti-acK311 for immunofluorescence applications.

Fig. 3e: The authors claim to have detected ac-K311 in both AD samples, however, in my opinion, the signal in AD#1 is too low to make this statement. In addition, in AD #2 I only see the lower band (5) and maybe one above but not on the same levels as the one in PiD #1. Have the authors quantified the data?

To evaluate whether all claims are convincing and the data is sound the authors should provide more information regarding the following points:

Results of mock transfections and mock-IPs are missing and should be added to the figures to provide controls for unspecific binding of the protein of interest.

Throughout the whole manuscript numbers of biological replicates are missing.

In general, the description of material and methods is not providing enough detail to properly evaluate and reproduce the data. The authors should check the methods section again and provide more details or cite according literature. Examples of missing information are i) dilutions of

antibodies (and for which experiment which antibody was used), ii) ages and number of animals, iii) detailed description of quantification of immunoblots (including normalization to loading controls or else).

The authors should provide the reference to the responsible committee and according regulations regarding animal behaviour experiments and crossing of mouse lines (page 26).

The authors claim that their "data warrant a careful evaluation of whether targeting, depleting or inactivating HDAC6 will provide therapeutic utility in ... tauopathies" (page 17, lines 15-16). However, a corresponding answer, statement or comment is missing and should be added to the discussion.

In addition, the authors should critically evaluate their conclusions again, e.g. on page 18 they suggest HDAC6 as an early AD biomarker. However, data from HDAC6-KO mice clearly show that the depletion of HDAC6 alone is not enough to develop neurodegenerative phenotypes. It is the combination of tauopathy and HDAC6-KO that leads to increased neurodegenerative phenotypes. Hence, HDAC6 alone would not be a suitable early biomarker.

Minor comments:

In general, the manuscript is well written, however, the authors could include a concrete research question in the introduction and a short paragraph with a summary of the main findings and conclusion at the end of the discussion.

In the whole study the authors used a variety of tau antibodies, several of them detecting "normal" tau. The authors should briefly explain the differences of those antibodies.

Page 3, line 19: "acetylated tau is undetectable in cognitively normal control brains, suggesting ac-K280 is a disease-specific marker for tauopathies": reference is missing.

Page 4, line 6/7: "Selective HDAC6 inhibition has shown varying degrees of neuroprotection in some AD models". According to the abstracts of references 16 & 17, these studies do not refer to models of AD but to tauopathies affecting oligodendrocytes. The authors should correct this.

Fig 1b-c: The authors mention that depletion of the C-terminal domain (HDAC6 1-840) or just the deletion of the minimal SE14 domain reduced tau binding "without affecting total tau levels". This is not true for HDAC6 1-840 as depicted in Fig. 1b. In regard to this, the authors have to clarify which protein levels were used to normalize the data. Less binding of tau in cells transfected with HDAC6 1-840 FL might be due to low abundance of tau in the input. Do you have an explanation for this reduced amount of tau in the input?

The resolution of Fig. 1c, e and m is very low.

Fig. 1j: The design of this graph is irritating because higher HDAC6 activity is depicted as 0 and low HDAC6 activity as 1 (see TSA). In addition, it is not clear to me whether in the positive control TSA was added to cells without any recombinant tau protein. If so, the authors also have to measure HDAC6 activity in cells without tau protein and include it in the panel as a "negative" control.

Fig. 1k: In the corresponding text the authors write "we note that HSP70, but not Hsc70, enhanced tau clearance". Do the authors refer here to the decreased levels of tau in the input? If yes, the authors should clarify this in the text. In addition, does this mean that the authors transfected the cells with HSP-constructs prior to Co-IP? If so, they should mention this in the methods section and add information about the constructs.

Figure 2 is confusing and needs more explanation in the figure legend and/or the corresponding text in the results section. If the idea was to show a robust co-aggregation of HDAC6 and tau by showing the same effect in different model systems and for different tau forms (P301L, P301S, Tau-1, normal tau) the authors should state this clearly in the text.

Fig. 2a: It should read Tau P301L-GFP instead of Tau-GFP. The authors claim that there is "extensive co-localization" of HDAC6-FL and Tau-P301L in cortical neurons of Tau-KO mice. However, both signals are overexposed, which makes it hard to follow this statement. In addition, it is not clear whether the pictures are projections of confocal z-stacks or single confocal planes.

Fig. 2b: Have the authors tested specificity of the used HDAC6 antibody? I suggest to conduct the same experiment using neurons from PHK mice to compare the results and check the antibody for specificity as the signal should be gone in PHK neurons.

Fig. 2d-e: It should read insoluble instead of insolubule.

Fig. 2f: In the results section the authors should make clear that in Fig. 2f, AD cortical neurons from human were analyzed. In addition, they should explain why they compared conformational MC-1 tau and TAU-5.

Fig. 3a-b: Quantification is questionable since there is no detectable signal in control samples. The authors should critically rethink their way of quantifying their immunoblot data. In addition, the authors should unify the y-axes in 3b.

Fig. 4a: The authors should give more detail in regard to their experimental design. Were the cells transfected with Tau, CBP and/or HDAC6 at the same time? For how long?

Fig. 4b-d: The authors should explain what they mean with relative levels.

Fig. 4i: The resolution of is very low.

Fig 6: The authors should add corresponding data for HDAC6-KO animals to compare the effect of HDAC6-KO to a brain with tauopathy.

Fig. 6d: The authors should provide single channel images. In addition, the authors write in the results section that "PHK did not harbor more ThS-positive tau inclusions". To make this statement the authors should quantify their data.

Fig. 6e: Measurement of cortex and hippocampus atrophies have to be explained in more detail and representative images should be shown.

The resolution of Fig. 7h-i is very low.

Table 1: Information about statistical analysis is missing.

Suppl. Fig. 1: To study the critical domains of HDAC6 that mediate tau binding, the authors used a panel of HDAC6 mutants. The schematic figure (Suppl. 1b) is not very clear. I would suggest to add a figure with all WT and mutant HDAC6 constructs including the resulting number of amino acids (which would also make the immunoblots easier to read).

Suppl. Fig. 2 / Page 10: The authors should clarify the difference between oligomerization and aggregation of tau and clearly explain which methods were used for which question.

Suppl. Fig 3: As shown in Sup. Fig. 3b, the antibody which was chosen for this study, also showed a signal in the K311R mutant control. The authors should mention and discuss this briefly in the results section instead of saying that the antibody showed a "high sensitivity and specificity"

Suppl. Fig. 4: Scale bar is missing

Suppl. Table 2: The authors should make clear which cases were used for which experiment (biochemical, IF and IHC). Page 12, line 5ff: The authors explain that they have screened 42 PSP/CBD tauopathy cases. Did they screen for acK311 immunoreactivity? If yes, they should make this clear and give more details about the screening, e.g. to what was acK311 immunoreactivity compared (to control brains?), how was "rare" and "mild" quantified?

Reviewer #3 (Remarks to the Author):

The work by Trzeciakiewicz et al. is a well written and exciting study that provides strong experimental data supporting a critical role for tau acetylation in mediating altered function and the conclusion that the class IIb lysine deacetylase (HDAC6) plays a protective role in the presence of tau pathology.

Recent studies from the authors' and other laboratories have highlighted a key role of the acetylation of tau lysine side chains in modulating tau proteostasis, yet the relevance of these findings to being a driver of disease pathophysiology or a protective factor has remained unclear. Dissecting this contribution has been challenging at least in part due to the multiple lysine side chains in tau that are available for modification and the potential competing roles that different sites of modification might have. Moreover, conflicting evidence exists for whether HDAC6 inhibition is protective or detrimental in the context of human disease, with HDAC6 inhibition being the focus of pre-clinical investigation for a range of disorders outside of oncology where inhibitors were first examined, including in Alzheimer's disease, amyotrophic lateral sclerosis, neurodevelopment disorders, and peripheral neuropathy.

The authors begin their study by showing that HDAC6 binds to tau in a manner that is independent of the ubiquitin binding domain but appears to be mediated through the c-terminal region involving SE14 domain. They also show that HDAC6 binds tau in a complex with chaperone proteins thereby linking HDAC6 and tau to critical regulators of protein quality control. The authors further substantiate previous findings looking at HDAC6 localizing to sites of neuritic lesions and extend these observations to look in post-mortem human brain from Alzheimer's patients. Using further fraction of soluble and insoluble fractions of protein from Alzheimer's brain, HDAC6 is shown to be depleted from the soluble fraction and accumulate along with pathological tau species in a small set of Alzheimer brain samples. Using mass spectrometry, novel tau acetylation sites were then identified, including that of Lys311 in R3 regions that appears in a homologous region to the more well studied Lys280 on the R2 region, AcLys311 is shown to be a specific marker of 3R and 3R/4R tauopathies, which as the authors point out is surprising given the fact that AcLys280 is most prevalent in 4R tauopathies and that the R3 region is found in all 6 tau splice forms. The authors then link HDAC6 to Lys311 acetylation state through a series of functional cellular and biochemical studies.

Importantly, the effect of Hdac6 loss-of-function in the context of tau transgenic mice is looked at for the first time and the authors observed reduced survival, accelerated cognitive decline, and greater tau pathology. On the basis of these findings, the authors conclude that HDAC6 normally functions to suppress accumulation of toxic forms of tau. These results are significant given that previous studies using the small molecule HDAC6 inhibitor tubastatin A have suggested that acute HDAC6 inhibition ameliorates cognitive and behavioral deficits in both amyloid-beta and tauopathy mouse models. Moreover, while Hdac6 knockout mice have not been reported to not have cognitive deficits, unlike the present study of Trzeciakiewicz et al., previous studies did not examine the effect of Hdac6 genetic loss-of-function and thus chronic inhibition in the context of neurodegeneration models in particular ones in which tau is perturbed genetically.

The studies appear to have used appropriate number of biological replicates and statistical methods. In most cases, the experiments appear well controlled. My recommendations for strengthening the manuscript are as follows:

- 1) In the co-immunoprecipitation experiments shown in Figure 1a, data are missing for HDAC1 and HDAC2, both of which are prominent class I HDACs in the CNS. Were these tested for interaction or found not to interact? This is of relevance given the fact that the authors go on latter to

implicate HDAC6-independent effects of HDAC inhibitor TSA treatment.

2) In Figure 1b a mutant of both of HDAC6's deacetylase domains is used. See in Fig 1 b elevated binding with (HDAC6 CD) and this is when to elevate the level of interaction of tau. Since this suggests that the inhibition of the deacetylase activity of HDAC6 stabilized the interaction with tau, it would be of great interest if the authors further dissected if loss-of-function of the N- or C-terminal deacetylase domain is sufficient for this stabilization or if mutations in both domains are needed.

3) The authors should clarify more of the technical details regarding the in vitro deacetylase assay conditions used to assess HDAC6 activity, including the nature of the substrate (since there are many different peptides used in this assay, and the relative concentration of HDAC6 to tau, the concentration of TSA used to set 100% inhibition, and demonstrate that the forms of tau tested do not inhibit trypsin activity alone that is necessary for a signal in this assay. In addition, since HDAC6 has two deacetylase domains, which domain (or domains) are being assayed?

4) Given that different HDAC6 pharmacological inhibitors may have different propensity to inhibit each of the two HDAC6 deacetylase domains, and that this may be important for efficacy and explain differences between different agents (e.g. TSA vs. tubastatin A), this raises the question of which of the 2 deacetylase domains mediates tau deacetylation and the relationship between HDAC6 inhibition by tau and this deacetylation event.

5) In Figure 1j, the authors see that P301L tau and S320F tau inhibits more than WT tau but that L315R tau, which had reduced HDAC6 binding inhibited HDAC6 just as well. The authors should explain the basis of inhibition being observed.

6) To understand the specificity of the HDAC6 inhibition being observed, the same forms of tau tested against HDAC6 should be tested against a class II HDAC (e.g. HDAC5) not found in the co-IP studies.

7) Since the P301S mutation is located in the Exon10 region and is thus a 4R tau specific mutation, it is unclear why the authors picked this specific mutation for the mouse model given their human post-mortem data suggesting that the elevation of of AcK311 levels was not found in 4R-specific tauopathies. To help link these studies together, the authors should include western blot data or ideally IHC data on a brain sample from a P301S mutation carrier.

8) In Figure 4F, it appears that there may be a decrease of total human tau as measured by Tau 12 at the 5 uM dose, but not lower doses.

Since the question of whether HDAC6 pharmacological inhibitors impact levels of tau are of great importance to clarify, the authors should separately plot out total tau levels (K9JA and Tau12) for both mouse (when possible) and human as well as total (when possible) as a function of Tubastatin A treatment.

9) Missing from Figure 4F is any evidence of HDAC6 target engagement, and therefore the addition of a panel with Acetyl-Lys40 alpha-tubulin relative to total tubulin should be included. Similarly, the level of AcK311 and AcK280 tau should be looked at in this full dose response and not just the single concentration shown in Fig. 4m

10) Tubastatin has a limited degree of selectivity for HDAC6 over class I HDACs. At the cellular concentrations tested in the 1-5 uM range, which is well above the IC50 for HDAC6 inhibition, the authors should determine if effects on nuclear class I HDACs leading to histone acetylation changes occur and ideally should add the testing of a non-HDAC6 targeting inhibitor that targets class I HDACs (e.g. MS-275) to determine if this also affects tau phosphorylation differences observed in Figure 4.

11) Unlike the behavioral characterization in Figure 5, the immunohistochemistry and biochemistry showed in Figure 6 and Figure 7 lacks the comparison to the effect of Hdac6-KO alone making the complete comparison of phenotypes problematic.

12) The cited studies by Selencia et al. (*Alzheimer's research & Therapy*, 2014) using the same compound of tubastatin administered daily for 2 months to tg4510 mice carrying a P301L mutation showed that tubastatin treatment rescued a memory deficit and normalized a hyperactivity phenotype in a manner correlated with lower total tau. While not the same mouse model and not the same behavioral tests, the results of this pharmacological experiment are in stark contrast to those presented by Trzeciakiewicz et al. showing an accelerated tauopathy phenotype. Thus, a key unresolved question from this study is whether repeated, long-term (> 2 months) dosing with a brain penetrant HDAC6 inhibitor in the background of a tau mutation has the same effects as the HDAC6 knockout does in terms of exacerbating the cognitive deficits and tau pathology. Given this divergent result and a desire to reconcile the findings, the authors should administer tubastatin for at least a 2 month time period in PS19 mice to determine if the differences are strain and time dependent.

Minor Points

13) Page 4, The authors should further clarify that so-called broad spectrum HDAC inhibitors, including those that are currently FDA approved, are inhibitors of class I HDACs (HDAC1/2/3/8) and class IIb (HDAC6/ HDAC10) but generally lack the ability to inhibitor class IIa HDACs (HDAC4/5/7/9). This includes the compound trichostatin A used in these studies.

14) Many HDACs show compensatory changes when genetically targeted. With the knockout of Hdac6, does the level of the other class IIb HDAC10 change?

Below, we respond to the reviewer's comments and describe revisions that we have done to address their concerns. Their comments are written in black and our responses are written in blue.

Reviewer #1 comments:

Overall an interesting study which I believe will be of interest to the field. The manuscript is written well and is nicely presented. For publication, however, several claims need to be strengthened with additional experiments and/or appropriate quantification.

1.1 In figure 1 the authors mention 'binding affinity' several times yet only co-immunoprecipitation has been conducted (page 5 line 12; page 6 line 1; page 6 line 11, page 6 line 17; page 6 line 23; page 7 line 4; page 7 line 15; page 7 line 22; page 7 line 23). Co-IP cannot determine the affinity between two proteins and if the authors wish to make claims regarding affinity or whether the binding between two proteins is reduced or increased, then surface plasmon resonance needs to be conducted (or the Blitz system utilized).

We have now changed this wording to "binding" in all instances. Co-IP assays are a well-accepted assay to determine interactions between two proteins.

1.2. Were the images in Figure 2 captured using a confocal microscope with z-stacks? If only a fluorescence microscope was used then the term co-localisation should not be used

Images in Fig. 2a-b were taken on the Olympus IX83 inverted microscope and confirmed by confocal imaging with z-stacking on a Zeiss 780 microscope at our UNC microscopy core (Supplementary Fig. 2). Co-localization of tau and HDAC6 in neuritic AD plaques from human samples was performed on the same Zeiss LSM 780 confocal microscope (Fig. 2f). In addition, we now provide an orthogonal view of the co-localization for final confirmation (Supplementary Fig. 2b).

1.3. Page 13 line 3: the authors mention that S202 and T205 p-epitopes don't increase yet in Figure 4d there is a significant increase in these levels

The quantification of the immunoblots from Fig. 4a has now been re-analyzed per reviewer's comment in 1.5 and normalized to total tau (Tau 12) rather than GAPDH (Fig 4b-d). In the presence of CBP, we observe increases in tau phosphorylation at S262, S356, S202, and S404. However, upon adding HDAC6, only phosphorylation within the MTBR was reduced (S262 and S356) while other sites flanking the MTBR were not dephosphorylated by the presence of HDAC6 (Fig. 4c-d).

1.4. Page 13 line 4: the authors mention that there is reduced tau phosphorylation at MARK2 sites S262 and S356 but S396 is also reduced in figure 4d

See comment 1.3 above. New quantification graphs have been provided in Fig. 4b-d with the re-analyzed data normalized to total tau (Tau 12). Only the MTBR phosphorylation sites S262 and S356 exhibited reduced phosphorylation by HDAC6 (compare Tau+CBP with Tau+CBP+HDAC6).

1.5. For Figure 4, what were acetylated and p-tau levels normalized to? If acetylation of tau increases the half-life of tau then post-translationally modified tau levels should be normalized to total tau rather than a housekeeping protein

Acetylated tau and phosphorylated tau in Fig 4b-d have now been normalized to total tau (Tau 12) and not the housekeeping GAPDH protein.

1.6 Figure 6a-d: quantification needs to be conducted to support claims

We have now added new data showing quantification of MC1, AT8, and Thioflavin-S (ThS) staining in Fig. 6e.

1.7 Figure 6e: can you please include the images for this data

We have now added representative images used for the hippocampal atrophy measurements in a new figure, see Supplementary Fig. 9b.

1.8. Figure 7a-b: please see comment 6 above

The immunoblots depicted in Fig. 7 are quantified in Fig 7f-i.

1.9. Page 17 line 20: “ We now demonstrate significantly increased acetylated tau in tauopathy mice lacking HDAC6....” Please refer to comment 6

All acetylated tau immunoblots soluble and insoluble are quantified in Figure 7f.

1.10 Page 19 line 18,”.... Increased affinity to P301L and S320F mutations.” Needs to verified with additional experiments.

We have now changed the wording to reflect increased binding rather than affinity as addressed in reviewer’s comment 1.1.

1.11 Unpaired t-tests were used frequently for statistical analysis of more than two groups- please repeat statistical analysis of these groups with a one-way ANOVA

In Figures 1-4, these experiments require statistical analysis between two groups (e.g. control vs. experimental group) and therefore t-tests were the most appropriate analysis. Furthermore, in all comparisons amongst multiple genotypes, we now report the data with one-way ANOVAs and Tukey’s post-hoc analysis tests for multiple comparisons among genotypes (Figure 6f, and Figure 7f-i). Mouse behavioral data was already reported as ANOVAs with Fischer’s protected least-significant difference test. The statistical analysis section of the methods has now been updated to address the reviewer’s comment.

Reviewer #2 comments:

In a previous study, the authors of this manuscript have identified HDAC6 as a tau deacetylase and have shown that acetylation of tau inhibits its function and promotes pathological tau aggregation (Cohen et al. 2011, Nat. Comm.). In this study they investigated the role of HDAC6 in tauopathy progression. The first part deals with the molecular interplay between tau and HDAC6. Here, the authors identified a complex of HDAC6 and HSP70 targeting tau via the microtubule binding region in vitro and claimed that HDAC6 accumulates in pathological tau lesions in mice and human AD brain tissue. Using a new ac-K311 antibody the authors then show that hyperacetylation of tau is preferentially occurring in 3R-tauopathies compared to 4R-tauopathies. Finally, they show that there is an interplay between HDAC6-mediated deacetylation and the phosphorylation status of tau. In the second part of the manuscript, the authors have used mouse models to investigate the effect of HDAC6-KO on

tauopathies. Using a newly created mouse line called PHK (PS19 & HDAC6-KO), they observed a decline in cognitive functions as well as accelerated tau pathology and neurodegeneration in HDAC6 deficient mice (PHK) when compared to the tauopathy model (PS19).

With this study the authors tackle an ongoing and controversial question, namely “is HDAC6 activity protective or detrimental in neurodegenerative diseases?”. The assumption that depleting or inactivating HDAC6 activity might have detrimental effects is not new to the field (see review: Simoes-Pires et al. 2013, Mol. Neurodegen.). The authors of this study, however, add new knowledge about underlying mechanisms of the interplay between HDAC6 and tau in tauopathy models. The manuscript is generally well written, the choice of models and methods is appropriate and the claims of the study are novel and nicely build upon previous findings of the authors. However, the role of acetylated tau, including the interplay of tau and HDAC6, has been characterized in various studies before (see recent review: Naseri et al. 2019, Neurosc. Letters, 10.1016/j.neulet.2019.04.022). Hence, the authors should point out the novelty of their findings in regard to previous findings in more detail.

In summary, there are several weaknesses as pointed out below that need to be addressed before this manuscript can be recommended for publication in Nature Communications.

Major comments:

2.1 The authors call their manuscript “An HDAC6-dependent surveillance mechanism suppresses tau-mediated neurodegeneration and cognitive decline”. However, this title is misleading because what this study shows is that transgenic mice lacking HDAC6 showed accelerated tau pathology, including neurodegeneration and cognitive decline. Therefore, the title should be adapted.

Based on our data, our title is appropriate. We show that HDAC6 recognizes aberrant forms of tau (Fig. 1) and that loss of HDAC6 sensitizes mice to tauopathy (Figs. 5-7). This indicates that HDAC6 has a protective role in the brain to keep tau deacetylated, e.g. HDAC6 acts to survey the neuronal environment and prevent abnormal tau protein aggregation. We believe our findings justify the title.

2.2 A major claim of this study is that 3R tau is selectively acetylated. However, the data presented here is not strong enough to support this notion (see comments below), especially because the authors mention that in in vitro studies 3R and 4R tau isoforms are identically acetylated by CBP.

Allow me to clarify. A major claim of this study is that 3R tauopathies show selective tau acetylation at the K311 site, in contrast to 4R tauopathies. The in vitro HEK-293A cell culture model, where we show CBP is able to acetylate all tau isoforms 3R and 4R tau, is actually quite different and not reflective of disease pathology or disease progression (Supplementary Fig. 4c). Rather in human brain, we observe that tau forms distinct “strains” or conformational variants in AD and PiD brains (3R tauopathies) that are positive for ac-K311, whereas CBD and PSP brains (4R tauopathies) are negative. This data clearly show that there is a preference for 3R tau acetylation at residue K311 in vivo (Fig. 3c and Fig 3d). We even screened 42 additional 4R tauopathy brain cases by IHC in Supplementary Table 1 to confirm this notion. Therefore, we conclude that in vitro all tau isoforms have the potential to be acetylated at K311, however in a disease setting, K311 acetylation emerges in a strikingly different pattern depending on the particular tauopathy. We addressed this in the results on Page 14, Line 6-8.

2.3 Fig. 3c: In order to evaluate aberrant acetylation of tau in 3R and 4R tauopathies the authors should add a figure showing the level of acK311 in normal healthy brain.

The levels of ac-K311 in normal healthy brain are shown by immunoblots in Fig. 3a from control brains. Per reviewer’s request, we also provided new IHC data from control brains in Supplementary

Fig. 5a. Finally, see Supplementary Table 1 for a list all of control brains analyzed by IHC and negative for K311 acetylation.

2.4 Fig. 3d: The authors claim to show colocalization of Tau and ac-K311 in 3R tauopathies (AD and PiD), however also 4R-tau in AD colocalized with ac-K311. The authors should mention and discuss this. In the third panel (PiD – RD3) both, tau and acK311 signals are highly overexposed. The authors should use different images to show that these staining are not artifacts. Also, the authors should describe how they have tested the specificity of anti-acK311 for immunofluorescence applications.

While 4R-tau did co-localize with ac-K311 in AD brains, we note that the extent of co-localization was not as robust as 3R-tau, which we have now commented on in the results section, on Page 13, Line 5-7.

The images that the reviewer mentions have now been adjusted. Non-diseased control brains were always negative for the ac-K311 antibody providing the appropriate negative control (Supplementary Table 1). Single-channel labeling was performed and does not show any evidence of cross-channel bleed-through, further suggesting that this is not a staining artifact. Pick bodies are very dense abundant structures, which is reflected in our images.

2.5 Fig. 3e: The authors claim to have detected ac-K311 in both AD samples, however, in my opinion, the signal in AD#1 is too low to make this statement. In addition, in AD #2 I only see the lower band (5) and maybe one above but not on the same levels as the one in PiD #1. Have the authors quantified the data?

The samples in Fig. 3e are purified PHF-tau from human brain cases. Therefore, the detection sensitivity is low in these biochemical brain fractions. We showed that ac-K311 detected smeared tau aggregates in insoluble AD cases (Fig. 3a), which likely represent multimeric tau species (i.e. oligomers) that are not present in the isolated PHF tau preparations. Nonetheless, ac-K311 was still able to detect monomeric tau pathology in PiD. This experiment is strictly a qualitative assessment of tau protein banding pattern to confirm the presence of ac-K311 tau in AD and PiD, but less so in CBD and PSP. A much larger panel of cases were analyzed and reported in Supplementary Table 1.

To evaluate whether all claims are convincing and the data is sound the authors should provide more information regarding the following points:

2.6 Results of mock transfections and mock-IPs are missing and should be added to the figures to provide controls for unspecific binding of the protein of interest.

Mock transfections and mock-IPs have now been included. For example, mock-IPs containing Tau with a control empty vector (pCDNA3.1 vector lacking a deacetylase) are found in Supplementary Fig. 1c-d. Mock-IPs with control empty vector (pCDNA3.1 vector without a Hsp) are shown in Fig. 1k.

The ideal negative controls that the reviewer may also be referring to are tau mutants that no longer bind HDAC6. Indeed, we show that tau lacking R2 or R3 domains abrogates binding suggesting specific binding. Similarly, we show that HDAC6 mutants lacking the SE14 domain also reduce binding (Fig. 1). Therefore, these appropriate negative controls strongly suggest that the binding is indeed due to specific domains in each of the proteins.

2.7 Throughout the whole manuscript numbers of biological replicates are missing.

These have now been added throughout the manuscript including the figure legends and methods section under “statistical analysis” on Page 32-33.

2.8 In general, the description of material and methods is not providing enough detail to properly evaluate and reproduce the data. The authors should check the methods section again and provide more details or cite according literature. Examples of missing information are i) dilutions of antibodies (and for which experiment which antibody was used), ii) ages and number of animals, iii) detailed description of quantification of immunoblots (including normalization to loading controls or else). The authors should provide the reference to the responsible committee and according regulations regarding animal behaviour experiments and crossing of mouse lines (page 26).

The following adjustments have been made:

i) Dilutions of antibodies and for which experiments have now been added to Supplementary Table 2
ii) Ages and number of animals have now been added to the methods section and figure legends.
iii) Descriptions of quantification of immunoblots (including normalization to loading controls or else) has now been added to the figure legends and methods section under “immunoblotting and biochemical methods” on Page 26.

The responsible IACUC committee at UNC has now been properly acknowledged in the methods section.

2.9 The authors claim that their “data warrant a careful evaluation of whether targeting, depleting or inactivating HDAC6 will provide therapeutic utility in tauopathies” (page 17, lines 15-16). However, a corresponding answer, statement or comment is missing and should be added to the discussion.

We have now provided a follow-up statement in the discussion on Page 20, Line 2-5.

2.10 In addition, the authors should critically evaluate their conclusions again, e.g. on page 18 they suggest HDAC6 as an early AD biomarker. However, data from HDAC6-KO mice clearly show that the depletion of HDAC6 alone is not enough to develop neurodegenerative phenotypes. It is the combination of tauopathy and HDAC6-KO that leads to increased neurodegenerative phenotypes. Hence, HDAC6 alone would not be a suitable early biomarker.

We agree with the reviewer that our data suggest that HDAC6 is a biomarker in the context of tauopathy. Therefore, we have now re-phrased this comment on Page 20, Line 13-15.

Minor comments:

2.11 In general, the manuscript is well written, however, the authors could include a concrete research question in the introduction and a short paragraph with a summary of the main findings and conclusion at the end of the discussion.

We have now added a statement to the introduction on Page 5 and a summary paragraph to end of the discussion on Page 24.

2.12 In the whole study the authors used a variety of tau antibodies, several of them detecting “normal” tau. The authors should briefly explain the differences of those antibodies.

All of the tau antibodies used have been well characterized in prior publications and are commercially available with detailed specification sheets. We used K9JA and TAU-5 total tau antibodies, which are commonly used to detect total mouse and human tau. We also used Tau 12, which is commonly used to detect total human tau. Supplementary Table 2 lists all tau antibodies used within our study. Any new antibodies generated in this study (e.g. multiple versions of ac-K311) have been documented and characterized with appropriate controls (Supplementary Fig. 4), and are available upon request.

2.13 Page 3, line 19: “acetylated tau is undetectable in cognitively normal control brains, suggesting ac-K280 is a disease-specific marker for tauopathies”: reference is missing.

This reference has now been added.

2.14 Page 4, line 6/7: “Selective HDAC6 inhibition has shown varying degrees of neuroprotection in some AD models”. According to the abstracts of references 16 & 17, these studies do not refer to models of AD but to tauopathies affecting oligodendrocytes. The authors should correct this.

These references have been corrected and the wording has now been re-phrased to refer to “tauopathies”.

2.15 Fig 1b-c: The authors mention that depletion of the C-terminal domain (HDAC6 1-840) or just the deletion of the minimal SE14 domain reduced tau binding “without affecting total tau levels”. This is not true for HDAC6 1-840 as depicted in Fig. 1b. In regard to this, the authors have to clarify which protein levels were used to normalize the data. Less binding of tau in cells transfected with HDAC6 1-840 FL might be due to low abundance of tau in the input. Do you have an explanation for this reduced amount of tau in the input?

Total tau levels were not statistically different among samples, please see immunoblot in Fig. 1b, input, total tau (K9JA). Furthermore, the data quantified in Fig. 1c (Fold HDAC6 Binding) is normalized to the total tau input. Therefore, we conclude that there is not a significantly reduced amount of tau in the input that would alter the fold HDAC6 binding, explained on Page 6, Line 6-9.

2.16 The resolution of Fig. 1c, e and m is very low.

This has been adjusted.

2.17 Fig. 1j: The design of this graph is irritating because higher HDAC6 activity is depicted as 0 and low HDAC6 activity as 1 (see TSA). In addition, it is not clear to me whether in the positive control TSA was added to cells without any recombinant tau protein. If so, the authors also have to measure HDAC6 activity in cells without tau protein and include it in the panel as a “negative” control.

The graph has now been re-analyzed and further clarified in the methods section on Page 31. We note that this assay was performed (without cells) in vitro using recombinant purified tau and HDAC6 protein, and a fluorescent FLUOR-DE-LYS reporter as a readout of HDAC6 activity. We incorporated TSA as a positive control for HDAC6 inhibition, to which all samples are normalized (Enzo life science).

2.18 Fig. 1k: In the corresponding text the authors write “we note that HSP70, but not Hsc70, enhanced tau clearance”. Do the authors refer here to the decreased levels of tau in the input? If yes, the authors should clarify this in the text. In addition, does this mean that the authors transfected the cells with HSP-constructs prior to Co-IP? If so, they should mention this in the methods section and add information about the constructs.

We have now clarified the reduction of total tau in the input, see Page 8, Line 9-11. As the reviewer indicated, cells were co-transfected with tau and the individual Hsp mentioned, the Hsp was pulled down and then blotted for tau. This point has now been clarified in the figure legend.

2.19 Figure 2 is confusing and needs more explanation in the figure legend and/or the corresponding text in the results section. If the idea was to show a robust co-aggregation of HDAC6 and tau by

showing the same effect in different model systems and for different tau forms (P301L, P301S, Tau-1, normal tau) the authors should state this clearly in the text.

We have now clearly stated this point in the results section, as the reviewer rightly suggested. On page 9, we now state, “To examine tau and HDAC6 localization and potential co-aggregation in neurons...”

2.20 Fig. 2a: It should read Tau P301L-GFP instead of Tau-GFP. The authors claim that there is “extensive co-localization” of HDAC6-FL and Tau-P301L in cortical neurons of Tau-KO mice. However, both signals are overexposed, which makes it hard to follow this statement. In addition, it is not clear whether the pictures are projections of confocal z-stacks or single confocal planes.

The labeling now reads as “P301L Tau-GFP”. We note that the fluorescent signals were not overexposed, as they are in the linear range of the microscope and the signals also co-localize by confocal imaging, confirming their association in Supplementary Fig. 2.

2.21 Fig. 2b: Have the authors tested specificity of the used HDAC6 antibody? I suggest to conduct the same experiment using neurons from PHK mice to compare the results and check the antibody for specificity as the signal should be gone in PHK neurons.

As the authors suggested, please see the HDAC6 immunoblot in Fig. 7a, in which the HDAC6 signal is eliminated in PHK mice lacking HDAC6 protein. Also, the specificity of the HDAC6 antibody used was further confirmed in the HDAC6 KO mice in Supplementary Fig. 7.

2.22 Fig. 2d-e: It should read insoluble instead of insolubule.

This has been updated.

2.23 Fig. 2f: In the results section the authors should make clear that in Fig. 2f, AD cortical neurons from human were analyzed. In addition, they should explain why they compared conformational MC-1 tau and TAU-5.

These points have now been clarified in the figure legends and main text on Page 10, Line 7-10. TAU-5 and MC1 antibodies are commonly used in the field to mark total tau or conformationally altered tau, respectively, present within tau tangles.

2.24 Fig. 3a-b: Quantification is questionable since there is no detectable signal in control samples. The authors should critically rethink their way of quantifying their immunoblot data. In addition, the authors should unify the y-axes in 3b.

The quantification has now been normalized to the more abundant AD samples, which were set to 100%. The y-axis has now been unified.

2.25 Fig. 4a: The authors should give more detail in regard to their experimental design. Were the cells transfected with Tau, CBP and/or HDAC6 at the same time? For how long?

These technical details have now been updated in the figure legends including Fig. 4a.

2.26 Fig. 4b-d: The authors should explain what they mean with relative levels.

The y-axis has now labeled as “Relative Levels (Normalized to Tau 12)”, to clearly state that the signal is reported as a ratio of modified tau to total tau and is also explained in the figure legends.

2.27 Fig. 4i: The resolution of is very low.

This has been adjusted.

2.28 Fig 6: The authors should add corresponding data for HDAC6-KO animals to compare the effect of HDAC6-KO to a brain with tauopathy.

We note that HDAC6 KO mice show no pathology, develop normally, and show no neurodegenerative phenotypes. This has been well documented in several previous studies (we have now provided the pertinent references for these studies in the results section on Page 17 Line 4-6). Nonetheless, we have now added new data to the revised manuscript showing no changes in tau pathology (Supplementary Fig. 7). Therefore, this genotype was removed from the analysis in Figure 6-7, since these mice are indistinguishable from WT mice by all measures.

2.29 Fig. 6d: The authors should provide single channel images. In addition, the authors write in the results section that “PHK did not harbor more ThS-positive tau inclusions”. To make this statement the authors should quantify their data.

We now provide single channel images in Supplementary Fig. 9a and new quantification of ThS pathology in Fig. 6e.

2.30 Fig. 6e: Measurement of cortex and hippocampus atrophies have to be explained in more detail and representative images should be shown.

We now provide representative images in Supplementary Fig. 9b and have further elaborated in the methods how we measured hippocampal atrophy on Page 32, Line 46 to Page 33, Line 1-2.

2.31 The resolution of Fig. 7h-i is very low.

This has been adjusted.

2.32 Table 1: Information about statistical analysis is missing.

The Table 1 legend contains the required $-10\log P$ (the “P” represents a protein confidence factor) and parts per million (ppm) measures the mass accuracy; both of these metrics confirm the statistical significance of the peptides identified by mass spectrometry.

2.33 Suppl. Fig. 1: To study the critical domains of HDAC6 that mediate tau binding, the authors used a panel of HDAC6 mutants. The schematic figure (Suppl. 1b) is not very clear. I would suggest to add a figure with all WT and mutant HDAC6 constructs including the resulting number of amino acids (which would also make the immunoblots easier to read).

Per the reviewer’s comment, the schematic in Supplementary Fig 1b has been updated to include the HDAC6 constructs used.

2.34 Suppl. Fig. 2 / Page 10: The authors should clarify the difference between oligomerization and aggregation of tau and clearly explain which methods were used for which question.

We have now updated the methods section to clearly describe that Thioflavin-T and pelleting assays are used to monitor fibrillar tau aggregates, while dynamic light scattering is used to detect multimeric tau species, or tau oligomers.

2.35 Suppl. Fig 3: As shown in Sup. Fig. 3b, the antibody which was chosen for this study, also showed a signal in the K311R mutant control. The authors should mention and discuss this briefly in the results section instead of saying that the antibody showed a “high sensitivity and specificity”

The wording in the results section has been updated to reflect minimal detection of an acetylation-deficient K311R mutation on Page 12, Line 5-9.

2.36 Suppl. Fig. 4: Scale bar is missing

The scale bar has now been included.

2.37 Suppl. Table 2: The authors should make clear which cases were used for which experiment (biochemical, IF and IHC). Page 12, line 5ff: The authors explain that they have screened 42 PSP/CBD tauopathy cases. Did they screen for acK311 immunoreactivity? If yes, they should make this clear and give more details about the screening, e.g. to what was acK311 immunoreactivity compared (to control brains?), how was “rare” and “mild” quantified?

We have now indicated which cases were used in each experiment. All human cases that were screened are now listed in Supplementary Table 1 with all pertinent details regarding demographics including whether ac-K311 immunoreactivity was positive or negative within pathological tau lesions.

Reviewer #3 comments:

The work by Trzeciakiewicz et al. is a well written and exciting study that provides strong experimental data supporting a critical role for tau acetylation in mediating altered function and the conclusion that the classIb lysine deacetylase (HDAC6) plays a protective role in the presence of tau pathology.

Recent studies from the authors' and other laboratories have highlighted a key role of the acetylation of tau lysine side chains in modulating tau proteostasis, yet the relevance of these findings to being a driver of disease pathophysiology or a protective factor has remained unclear. Dissecting this contribution has been challenging at least in part due to the multiple lysine side chains in tau that are available for modification and the potential competing roles that different sites of modification might have. Moreover, conflicting evidence exists for whether HDAC6 inhibition is protective or detrimental in the context of human disease, with HDAC6 inhibition being the focus of pre-clinical investigation for a range of disorders outside of oncology where inhibitors were first examined, including in Alzheimer's disease, amyotrophic lateral sclerosis, neurodevelopment disorders, and peripheral neuropathy.

The authors begin their study by showing that HDAC6 binds to tau in a manner that is independent of the ubiquitin binding domain but appears to be mediated through the c-terminal region involving SE14 domain. They also show that HDAC6 binds tau in a complex with chaperone proteins thereby linking HDAC6 and tau to critical regulators of protein quality control. The authors further substantiate previous findings looking at HDAC6 localizing to sites of neuritic lesions and extend these observations to look in post-mortem human brain from Alzheimer's patients. Using further fraction of soluble and insoluble fractions of protein from Alzheimer's brain, HDAC6 is shown to be depleted from the soluble fraction and accumulate along with pathological tau species in a small set of Alzheimer brain samples. Using mass spectrometry, novel tau acetylation sites were then identified, including that of Lys311 in R3 regions that appears in a homologous region to the more well studied Lys280 on the R2 region, AcLys311 is shown to be a specific marker of 3R and 3R/4R tauopathies, which as the authors point out is surprising given the fact that AcLys280 is most prevalent in 4R

tauopathies and that the R3 region is found in all 6 tau splice forms. The authors then link HDAC6 to Lys311 acetylation state through a series of functional cellular and biochemical studies.

Importantly, the effect of Hdac6 loss-of-function in the context of tau transgenic mice is looked at for the first time and the authors observed reduced survival, accelerated cognitive decline, and greater tau pathology. On the basis of these findings, the authors conclude that HDAC6 normally functions to suppress accumulation of toxic forms of tau. These results are significant given that previous studies using the small molecule HDAC6 inhibitor tubastatin A have suggested that acute HDAC6 inhibition ameliorates cognitive and behavioral deficits in both amyloid-beta and tauopathy mouse models. Moreover, while Hdac6 knockout mice have not been reported to not have cognitive deficits, unlike the present study of Trzeciakiewicz et al., previous studies did not examine the effect of Hdac6 genetic loss-of-function and thus chronic inhibition in the context of neurodegeneration models in particular ones in which tau is perturbed genetically.

The studies appear to have used appropriate number of biological replicates and statistical methods. In most cases, the experiments appear well controlled. My recommendations for strengthening the manuscript are as follows:

3.1 In the co-immunoprecipitation experiments shown in Figure 1a, **data are missing for HDAC1 and HDAC2**, both of which are prominent class I HDACs in the CNS. Were these tested for interaction or found not to interact? This is of relevance given the fact that the authors go on latter to implicate HDAC6-independent effects of HDAC inhibitor TSA treatment.

Based on the reviewer's comment, we have now tested HDAC1 binding to tau by co-IP assay and provide new data in Supplementary Fig. 1c showing minimal, if any, binding to this class I HDAC.

3.2 In Figure 1b a mutant of both of HDAC6's deacetylase domains is used See in Fig 1 b elevated binding with (HDAC6 CD) and this is is when to elevate the level of interaction of tau. Since this suggests that the inhibition of the deacetylase activity of HDAC6 stabilized the interaction with tau, it would be of great interest if the authors further dissected if loss-of-function of the N- or C-terminal deacetylase domain is sufficient for this stabilization or if mutations in both domains are needed.

Based on the reviewer's comment, we have now tested individual HDAC6 mutants in which either the CD1 or CD2 deacetylase domain has been inactivated by point mutations (H216A and H611A respectively). These new data are now shown in Supplementary Fig. 1d and Supplementary Fig. 6a. Based on our results, only the 2nd deacetylase domain is responsible for tau deacetylation, as the H611A mutant alone is sufficient to prevent tau deacetylation. We also observed increased binding of tau to the H611A mutant (Supplementary Fig. 1d), similar to that of H216/611A tau-binding in Fig. 1b, again suggesting that only CD2 is pertinent to targeting tau substrates.

3.3 The authors should clarify more of the technical details regarding the in vitro deacetylase assay conditions used to assess HDAC6 activity, including the nature of the substrate (since there are many different peptides used in this assay, and the relative concentration of HDAC6 to tau, the concentration of TSA used to set 100% inhibition, and demonstrate that the forms of tau tested do not inhibit trypsin activity alone that is necessary for a signal in this assay. In addition, since HDAC6 has two deacetylase domains, which domain (or domains) are being assayed?

The in vitro deacetylase assay was performed with purified recombinant tau and HDAC6 proteins in the presence of a fluorescent-based HDAC6 FLUOR-DE-LYS substrate reporter that is specific to HDAC6 but not other HDACs (the kit is commercially available from Enzo Life Sciences). We have now included the protein concentrations and additional details in the methods section on Page 31. In

addition, our new data indicate that only the CD2 domain is required for tau deacetylation (Supplementary Fig. 6a).

3.4 Given that different HDAC6 pharmacological inhibitors may have different propensity to inhibit each of the two HDAC6 deacetylase domains, and that this may be important for efficacy and explain differences between different agents (e.g. TSA vs. tubastatin A), this raises the question of which of the 2 deacetylase domains mediates tau deacetylation and the relationship between HDAC6 inhibition by tau and this deacetylation event.

We now provide new data showing that all of HDAC6's catalytic activity towards tau resides in the CD2, as an inactivating mutation in this domain (H611A) abrogated deacetylase activity, as shown in Supplementary Fig. 6a.

3.5 In Figure 1j, the authors see that P301L tau and S320F tau inhibits more than WT tau but that L315R tau, which had reduced HDAC6 binding inhibited HDAC6 just as well. The authors should explain the basis of inhibition being observed.

While P301L and S320F bound and inhibited HDAC6 more effectively, the L315R mutant was not statistically different compared to WT tau (Fig. 1j). We discuss this in the manuscript on Page 7, Line 13-16. We further suggest that in cells, HDAC6 binding is partly mediated by an HSP adaptor, namely Hsp70 or Hsc70, as we demonstrate in Fig. 1k, which could explain the slight differences in tau/HDAC6 binding observed in cells vs. in vitro using purified recombinant proteins.

3.6 To understand the specificity of the HDAC6 inhibition being observed, the same forms of tau tested against HDAC6 should be tested against a class II HDAC (e.g. HDAC5) not found in the co-IP studies.

In Supplementary Fig. 1e, we have now performed in vitro deacetylation assays with HDACs 1, 3, and 6 in parallel and we find that the patterns of HDAC inhibition by tau are very different between these three unique HDACs. While the negative control Δ R1-4 alleviated the HDAC6 inhibition, this mutant had no impact on HDAC1 or HDAC3. In addition, while P301L tau was more effective than WT tau at inhibiting HDAC6 activity, this was not the case for HDAC1 and HDAC3. Thus, the impact of tau on HDAC6 is quite specific. We discuss this on Page 7, Line 16-23.

3.7 Since the P301S mutation is located in the Exon10 region and is thus a 4R tau specific mutation, it is unclear why the authors picked this specific mutation for the mouse model given their human post-mortem data suggesting that the elevation of ac-K311 levels was not found in 4R-specific tauopathies. To help link these studies together, the authors should include western blot data or ideally IHC data on a brain sample from a P301S mutation carrier.

We have now included analysis of a MAPT human case containing the P301L mutation, which harbors predominantly 4R-tau (see Supplementary Fig. 5c). This case is ac-K311 negative consistent with ac-K311 detecting predominantly 3R-tau pathology.

While the PS19 mice express mutated 4R-tau, one might wonder why 4R-tau mice show ac-K311 signal while the 4R-tau human cases are negative. We note that it's only in mice lacking HDAC6 that we observe ac-K311 signal, not in the PS19 mice alone (see immunoblot for soluble ac-K311 in Fig. 7a, it is nearly undetectable). Since HDAC6 preferentially binds 4R-tau compared to 3R-tau (Fig. 1f), we suspect that HDAC6 can better engage the 4R-tau isoforms and hence keep them deacetylated in humans, while 3R-tau is less efficiently deacetylated and hence more 3R-tau becomes ac-K311 positive.

3.8 In Figure 4F, it appears that there may be a decrease of total human tau as measured by Tau 12 at the 5 uM dose, but not lower doses. Since the question of whether HDAC6 pharmacological inhibitors impact levels of tau are of great importance to clarify, **the authors should separately plot out total tau levels (K9JA and Tau12) for both mouse (when possible) and human** as well as total (when possible) as a function of Tubastatin A treatment.

Per reviewer's request, we have now separately plotted K9JA and Tau 12 levels as a function of TBST treatment in Fig. 4j for K9JA and Fig. 4k for Tau 12. We did not observe a statistically significant reduction of tau levels due to TBST treatment.

3.9 **Missing from Figure 4F is any evidence of HDAC6 target engagement**, and therefore the addition of a panel with Acetyl-Lys40 alpha-tubulin relative to total tubulin should be included. Similarly, the level of AcK311 and AcK280 tau should be looked at in this full dose response and not just the single concentration shown in Fig. 4m

We now show HDAC6 target engagement by analyzing ac-tubulin levels normalized to total tubulin, which are increased in response to all TBST doses (Fig. 4f and 4l). As described in our earlier publications (Cohen et al, Nat Com, 2011 and Cohen et al., NSMB, 2013), endogenous neuronal mouse tau from primary neurons is not readily aggregated into mature tau pathology, which is why we cannot detect acetylated tau forms in primary neurons. We discuss this on Page 15, Line 20-23 and Page 16, Line 1-2.

3.10 Tubastatin has a limited degree of selectivity for HDAC6 over class I HDACs. At the cellular concentrations tested in the 1-5 uM range, which is well above the IC50 for HDAC6 inhibition, the authors should determine if effects on nuclear class I HDACs leading to histone acetylation changes occur and ideally should add the testing of a **non-HDAC6 targeting inhibitor that targets class I HDACs (e.g. MS-275)** to determine if this also affects tau phosphorylation differences observed in Figure 4.

While we appreciate the reviewer's comment, our study is focused on HDAC6, as there is mounting evidence that cytoplasmic HDAC6 is highly relevant to AD onset or progression. Other HDACs, while interesting, are beyond the scope of this study, but we nonetheless showed that HDAC1 does not bind tau in Supplementary Fig. 1c. Indeed, HDACs 2/3 have both been linked to tau pathology in separate studies (Liu et al, Mol Ther. 2017 and Janczura et al., PNAS, 2018), and may in fact work in combination with HDAC6 (either directly or indirectly) to regulate tau phosphorylation, which may be why we observed TSA-sensitive control of tau acetylation. These other HDACs likely engage tau by different mechanisms than HDAC6 (as suggested by our in vitro data in Supplementary Fig.1e), which could be the topic of an exciting future study that links multiple HDACs to tau pathology, as we discuss on Page 20, Line 15-19.

3.11 Unlike the behavioral characterization in Figure 5, the immunohistochemistry and biochemistry showed in **Figure 6 and Figure 7 lacks the comparison to the effect of Hdac6-KO alone** making the complete comparison of phenotypes problematic.

HDAC6 KO mice show no behavioral phenotypes or tau pathology, which has been well documented. We have now added new immunoblotting data to support this claim in Supplementary Fig. 7. There is no obvious tau hyper-phosphorylation or insoluble tau aggregates present in these mice. See comment 2.28.

3.12 The cited studies by Selencia et al. (Alzheimer's research & Therapy, 2014) using the same compound of tubastatin administered daily for 2 months to tg4510 mice carrying a P301L mutation showed that tubastatin treatment rescued a memory deficit and normalized a hyperactivity phenotype

in a manner correlated with lower total tau. While not the same mouse model and not the same behavioral tests, the results of this pharmacological experiment are in stark contrast to those presented by Trzeciakiewicz et al. showing an accelerated tauopathy phenotype. Thus, a key unresolved question from this study is whether repeated, long-term (> 2 months) dosing with a brain penetrant HDAC6 inhibitor in the background of a tau mutation has the same effects as the HDAC6 knockout does in terms of exacerbating the cognitive deficits and tau pathology. Given this divergent result and a desire to reconcile the findings, the authors should administer tubastatin for at least a 2 month time period in PS19 mice to determine if the differences are strain and time dependent.

As the reviewer mentioned, the notion that HDAC6 inhibitors may be effective at reducing tau pathology is based on a study from Selenica et al (Alz Res Ther, 2014) where they treated Tg4510 mice with tubastatin A. This model (Tg4510) is now considered flawed by the AD research community and is no longer an ideal model for pre-clinical testing of any small molecule therapeutics. There are major concerns with the transgene insertion site, as recently reported from Karen Ashe's group in Nature Communications (Gamache et al, Nat Commun, 2019), in which the transgene altered multiple loci unrelated to tau. Please see the fallout from this very important study posted on Alzforum, <https://www.alzforum.org/news/research-news/gene-disruption-haunts-tau-mouse-knock-ins-look-promising>).

In addition to the model being problematic, tubastatin A is not sufficiently brain penetrant, nor is it highly water-soluble (requiring DMSO, ethanol or other toxic solvents), making it a very poor choice for long-term dosing in animals, particularly in animals that undergo slow progressive neurodegeneration. Therefore 2 months of tubastatin A treatment is simply not desirable and not likely to yield a meaningful outcome.

Lastly, we are cognizant that HDAC6 inhibitors may behave differently than HDAC6 KO. For example, HDAC6 may perform critical scaffolding functions that do not necessarily depend on its catalytic activity. Therefore, our observation that complete loss of HDAC6 accelerates tauopathy is not in direct contradiction with the Selenica et al study. This is an important distinction between the two approaches and must be considered.

Nonetheless, we have provided new data that partially address this reviewer's point. We performed behavior analysis of a different mouse strain containing A β plaques (5xFAD mice) but lacking HDAC6 (5xFAD-HDAC6 KO mice). To do this, we crossed 5xFAD mice harboring abundant plaques to the same HDAC6 KO mice used in our tauopathy analysis and performed cognitive tests. Indeed, consistent with the previous literature, loss of HDAC6 is protective in a plaque model, which starkly contrasts with our results in PS19 mice. Our data suggest that loss of HDAC6 is only detrimental specifically in a setting of tauopathy (PS19), while in contrast loss of HDAC6 improves cognition in the A β plaque model. This new data supports the hypothesis that HDAC6 acts to protect against tauopathy by keeping tau deacetylated, but does not target A β . Comparing these new genetic models lacking HDAC6 (in both PS19 and 5xFAD models) with pharmacological HDAC6 inhibitor data will be the topic of a future study, since this is well beyond the scope of the current study.

Minor Points

3.13 Page 4, The authors should further clarify that so-called broad spectrum HDAC inhibitors, including those that are currently FDA approved, are inhibitors of class I HDACs (HDAC1/2/3/8) and class IIb (HDAC6/ HDAC10) but generally lack the ability to inhibitor class IIa HDACs (HDAC4/5/7/9). This includes the compound trichostatin A used in these studies.

We have now added this information to the introduction on Page 4, Line 18-21.

3.14 Many HDACs show compensatory changes when genetically targeted. With the knockout of Hdac6, does the level of the other class IIb HDAC10 change?

In the absence of HDAC6, we confirmed that HDAC6 substrates such as acetyl-tubulin were increased (see ac-tubulin immunoblot in Figure 7a), suggesting any compensatory HDACs capable of targeting similar substrates are not increased in the absence of HDAC6. We have also analyzed HDAC10 levels and we do not observe any compensatory increases in HDAC10 protein in mouse brains lacking HDAC6. This new data has been included in the revised Supplementary Fig. 7.

REVIEWER COMMENTS

Reviewer #1 (Remarks to the Author):

In the revised manuscript the authors have addressed many of the comments raised by the reviewers. In doing so, however, a few more concerns have been raised that I believe need to be addressed to strengthen the manuscript.

Major comments:

The main weakness of the manuscript is characterization of the PHK mice and the following comments relate to this part of the manuscript.

1. The inclusion of the ac-K311 labeling of human 4R FTL D P301L brain tissue helps support the case that acetylation of residue 311 is specific to 3R tauopathies but it doesn't support the use of the PS19 mouse model for crossing with the HDAC6 knockout. The authors need to make it clear in the text why this mouse model was specifically chosen, despite expressing 4R tau and pathological mutation at a site which was shown early on in the manuscript to inhibit HDAC6 activity when mutated. An explanation as to why the PHK mice demonstrate an increase in ac-K311, despite expressing human 4R tau, and why knocking out HDAC6 in PS19 mice has an effect on tau pathology when the P301S mutation is likely to already inhibit HDAC6 activity, should be included.
2. As the main hypothesis of the study was that inhibition of HDAC6 enhances tau acetylation and pathology, why do the authors not show histological analysis of acetylated tau levels in the brains of WT, PS19 and PHK mice? Particularly as the antibody has been used for IHC earlier in the manuscript.
3. It is unclear in figures 7f-i what relative levels mean as it isn't mentioned in the text. Have the protein levels been normalized to total tau prior to normalizing to WT mice, and if so, which total tau? More information needs to be provided in the materials and methods regarding how this data was quantified.
4. How IHC data was analyzed needs to be included in the materials and methods. No reference to the results of the cortical staining in Figure 6 has been included in the text. There is also no quantification of this data.
5. To claim that the HDAC6 KO mice do not display any neurodegenerative phenotype, immunoblotting with antibodies specific to additional pathological tau species (rather than just AT8) is important, especially as earlier experiments suggest that the AT8 epitope may not be affected by HDAC6 activity.

Minor comments:

1. It would be helpful to the reader if the species specificity for the pan-tau antibodies was listed on the blots rather than labeling the blots with 'total' tau. For instance, in figure 4a the blot labeled 'Tau12 (total tau)' would be better labeled 'Tau12 (human tau)'. It would also be helpful if it was stated in the text why each antibody was used for each particular experiment. Particularly the use of Tau1 in Figure 2B as it is more specific to dephosphorylated tau.
2. The age of mice when behavior was conducted and when mice were sacrificed should be included in the methods section.
3. Figure 2F needs human control tissue.
4. The rationale for looking at mouse and human tau levels individually in figure 4 is not clear and

should be included in the text.

5. Some graphs display all data points whereas others don't. For transparency it would be best if all data points were displayed.

6. Page 17 line 2: the word 'directly' is used twice in the one sentence.

7. This manuscript would benefit from including a graphical summary.

Reviewer #2 (Remarks to the Author):

The authors have addressed all comments that were raised in a satisfactory way. They gave reasonable explanations, provided new data and added methodological information, which further strengthened the manuscript. I have no further queries regarding this manuscript and support its acceptance by Nature Communications.

Reviewer #3 (Remarks to the Author):

The authors are thanked for their thoughtful response to the issues raised. As a whole, they have suitably revised the manuscript to address the major technical and conceptual concerns noted with the initial submission.

As requested, they have added new experimental data defining the specificity of the interaction of tau with HDAC6 relative to at least one Class I HDAC member HDAC1. They have provided new data showing that only 1 of the 2 deacetylase domains of HDAC6 is involved in tau deacetylation with an elegant dissection using site-specific mutations (H216/611A) in the HDAC6 catalytic domain showing that only the C-terminal domain has tau deacetylase activity.

Additionally, the inclusion of new data from analysis of a MAPT human case with a P301L mutation showing that it is negative for Ack311 levels is a helpful clarification in terms of comparisons made to the PS19 mouse model that was shown to have Ack311 detectable but only upon knockout of HDAC6.

Although I had requested that the authors consider performing a long-term HDAC6 inhibitor treatment in PS19 mice to try to reconcile the observations of Selencia et al. using in Tg4510 mouse model the authors have cited recent findings pointing to limitation so the Tg4510 mouse as one possible explanation and described limitations with tubastatin A that makes the experiment challenging for long-term administration. They have now included new findings comparing the effects of HDAC6 knockout in a 5X FAD mutant background (5xFHK) and again see protection rather than an acceleration of tauopathy. Taken together, and in light of experimental limitations due to COVID-19, I no longer think this experiment holds sufficient merit to hold up the publication of this experiment.

Overall, the conclusion that the genetic loss of HDAC6 accelerates rather than alleviates tauopathy in mice is important. The authors have made a strong contribution to the field by their detailed and rigorously studies that ultimately inform current efforts focused on understanding the role of HDAC6 in protein quality control pathways and targeting HDAC6 therapeutically.

Minor points:

The substrate in the HDAC6 (BML-AK516) FLUOR DE LYS assay is in fact not specific for HDAC6 as stated in the response letter. As indicated by the manufacture

(<https://www.enzolifesciences.com/BML-AK516/fluor-de-lys-hdac6-fluorometric-drug-discovery-kit/>) the substrate used in this kit is based upon the FLUOR DE LYS SIRT1, Deacetylase Substrate (Prod. No. BML-KI177), which is derived from residues 379-382 of p53 (Arg-His-Lys-Lys(Ac)), and is known to be deacetylated by a wide range of HDACs and sirtuins. This does not affect the conclusion of the authors.

Figure 4 is extremely dense and difficult to read given the size of figures and graphs.

REVIEWER COMMENTS

Reviewer #1 (Remarks to the Author):

In the revised manuscript the authors have addressed many of the comments raised by the reviewers. In doing so, however, a few more concerns have been raised that I believe need to be addressed to strengthen the manuscript.

Major comments:

The main weakness of the manuscript is characterization of the PHK mice and the following comments relate to this part of the manuscript.

1. The inclusion of the ac-K311 labeling of human 4R FTL D P301L brain tissue helps support the case that acetylation of residue 311 is specific to 3R tauopathies but it doesn't support the use of the PS19 mouse model for crossing with the HDAC6 knockout. **The authors needs to make it clear in the text why this mouse model was specifically chosen**, despite expressing 4R tau and pathological mutation at a site which was shown early on in the manuscript to inhibit HDAC6 activity when mutated. **An explanation as to why the PHK mice demonstrate an increase in ac-K311, despite expressing human 4R tau, and why knocking out HDAC6 in PS19 mice has an affect on tau pathology** when the P301S mutation is likely to already inhibit HDAC6 activity, should be included.

PS19 mice are viewed as one of the gold standards in the field since they recapitulate tau pathology and many of the neurodegenerative phenotypes. This is why we crossed this model to HDAC6 KO mice. It is only when HDAC6 is deleted that the PS19 model shows acetylated tau accumulation (PS19 mice do not have much acetylated tau, Fig. 7a), supporting the notion that, at least in mice, HDAC6 normally acts to keep tau deacetylated as a protective mechanism. We have now included a comment in the discussion stating that PS19 mice have minimal K311-acetylated tau on their own (PG 24, lines 1-3); this is quite similar to the low/absent ac-K311 seen in human patients with the FTL D-tau P301L mutation. Therefore, in this regard, the mouse and human findings are in agreement.

Genetically knocking out HDAC6 in the PS19 model is expected to show maximal tau acetylation at sites targeted by HDAC6, and this is indeed what we observed in PHK mice. This high level of tau acetylation is unlikely to be achieved in the PS19 mice, in which any inhibition of HDAC6 (by P301S tau) could never match the complete loss of function achieved by genetic deletion of HDAC6. We would argue that this must be the case since PS19 mice alone show minimal acetylated tau throughout the brain (Fig. 7a).

2. As the main hypothesis of the study was that inhibition of HDAC6 enhances tau acetylation and pathology, **why do the authors not show histological analysis of acetylated tau levels in the brains of WT, PS19 and PHK mice?** Particularly as the antibody has been used for IHC earlier in the manuscript.

As with all AD mouse models, the biochemical properties of transgenic tau in the PS19 mice (only one human tau isoform expressing 1N4R-tau) is likely to be different from the human tau pathology in which all six tau isoforms are present in the brain. Because of these expected differences between mice and humans, we have been unable to detect acetylated tau by histology in mouse tissue sections after extensive troubleshooting. It is currently unclear why this is the case, but we suspect this is due to a requirement for more stringent detergent-based extraction since acetylated tau is, in general, quite aggregated and insoluble, as you can see in the human IHC analysis in Fig. 3c. Indeed, our biochemical approaches using very high salt concentrations or SDS extraction of brain

lysates is widely considered the most sensitive approach to enrich for insoluble tau. This worked much better and allowed us to readily detect acetylated tau by immunoblotting, which was quite clear and rigorous (Fig. 7a).

3. It is unclear in figures 7f-i what relative levels mean as it isn't mentioned in the text. Have the protein levels been normalized to total tau prior to normalizing to WT mice, and if so, which total tau? More information needs to be provided in the materials and methods regarding how this data was quantified.

Relative levels were defined in the figure legends for 7f-i and in the "Immunoblotting and biochemical methods" on PG 26 Line 15-18. All protein levels represent densitometry values that have been background subtracted and then normalized to PHK mice (for tau immunoblots) or to WT mice (for synaptic markers). The reason for this is that WT mice do not express human tau and therefore PHK values represent the maximum level set to 100% (or 1.0 in the graphs) for the tau analysis in Fig. 7f-j, while the synaptic degeneration observed in PS19/PHK mice was compared relative to WT; in this case WT was used as the maximum level and set to 100% (or 1.0 in the graphs) in Fig. 7i.

4. How IHC data was analyzed needs to be included in the materials and methods. No reference to the results of the cortical staining in Figure 6 has been included in the text. There is also no quantification of this data.

Methods and statistical analysis for IHC from Fig 6 has now been included in the materials and methods section on PG 29, Lines 5-8. The most robust changes in pathology and brain atrophy were observed in the hippocampus, which is widely accepted as the most highly susceptible region in the PS19 mouse line. Given the variability that can often be present in cortical measurements, we opted to focus our quantification on the hippocampus and referenced this in the text on PG 18, Lines 20-23.

5. To claim that the HDAC6 KO mice do not display any neurodegenerative phenotype, **immunoblotting with antibodies specific to additional pathological tau species (rather than just AT8) is important**, especially as earlier experiments suggest that the AT8 epitope may not be affected by HDAC6 activity.

We agree with the reviewer, which is why we have now performed additional phospho-tau immunoblotting and, more importantly, solubility analysis of WT vs. HDAC6 KO mice (see the revised Supplementary Fig. 7c). Our analysis of phospho-S262 shows no difference between WT and HDAC6 KO mice in either soluble or insoluble fractions. Given the differences in public opinion on which p-tau epitopes are considered most pathological (we originally had AT8 and now have additionally included p-S262), we also evaluated total tau species (detected with the K9JA antibody) in both soluble and insoluble fractions and detected no differences in tau levels in insoluble fractions of HDAC6 KO compared to WT mice, consistent with the literature showing HDAC6 KO mice are cognitively normal. Lastly, it is worth noting that mature tangle pathology typically requires a tau transgene to recapitulate tau aggregation and therefore any significant pathology is not expected in non-tau transgenic animals such as the HDAC6 KO mice.

Minor comments:

1. It would be helpful to the reader if the species specificity for the pan-tau antibodies was listed on the blots rather than labeling the blots with 'total' tau. For instance, in figure 4a the blot labeled 'Tau12 (total tau)' would be better labeled 'Tau12 (human tau)'. It would also be helpful if it was stated in the text why each antibody was used for each particular experiment. Particularly the use of Tau1 in Figure 2B as it more specific to dephosphorylated tau.

We prefer the labeling “total tau” since there is some discrepancy in the literature about whether some human tau-specific antibodies (e.g. Tau12, HT7, and T14 monoclonal antibodies) are completely negative for the detection of mouse tau. Supplementary Table 2 clearly summarizes the details for human or mouse reactivity for all antibodies used in our study. Justifying every tau antibody used in the text is redundant, and due to space limitations we provided the summary table that will quite easily help the reader. However, the use of Tau-1 in Fig. 2b was explained and referenced on PG 9, Lines 16-18.

2. The age of mice when behavior was conducted and when mice were sacrificed should be included in the methods section.

Our manuscript now states the ages in the figure captions and methods section as follows:

- Mouse behavior: PG 30, Lines 39-40, the mice used were 7–9 months old
- Primary neuron cultures: PG 25, Line 27, primary neuron cultures were derived from embryonic dissections of Tau KO, WT, or PS19 mice at E16
- Mouse perfusions, fixation, and tissue harvest for biochemical experiments (fractionation and dot blotting): PG 27, Lines 26-28, were performed in 12-month old mice

3. Figure 2F needs human control tissue.

In the absence of any AD pathology, all images are blank. This was detailed for all controls listed in Supplementary Table 1. Normal control brains show no plaque or tangle pathology.

4. The rationale for looking at mouse and human tau levels individually in figure 4 is not clear and should be included in the text.

When analyzing PS19 cultured primary neurons, it is important to assess the human tau levels and evaluate if mouse tau levels are auto-regulated in a compensatory manner due to over-expressed human tau. Human tau transgenes can impact the endogenous mouse tau levels. The text in the results now reads, “We analyzed both human transgenic tau and endogenous mouse tau to distinguish among the total tau species”. We described this on PG 15, Lines 15-17.

5. Some graphs display all data points whereas others don’t. For transparency it would be best if all data points were displayed.

We disagree and feel it is best to avoid overcrowding the many graphs in this manuscript, which makes them more visually appealing and interpretable to the reader. The number of experimental and biological replicates for each experiment is clearly stated in the text and figure legends. In instances where all data points were necessary, we provided this information.

6. Page 17 line 2: the word ‘directly’ is used twice in the one sentence.

Thank you, this has been fixed.

7. This manuscript would benefit from including a **graphical summary**.

We thank the reviewer for this wonderful suggestion. A graphical summary model has now been added to the revised manuscript as Supplementary Fig. 10 depicting both healthy and disease conditions.

Reviewer #2 (Remarks to the Author):

The authors have addressed all comments that were raised in a satisfactory way. They gave reasonable explanations, provided new data and added methodological information, which further strengthened the manuscript. **I have no further queries regarding this manuscript and support its acceptance by Nature Communications.**

Reviewer #3 (Remarks to the Author):

The authors are thanked for their thoughtful response to the issues raised. As a whole, they have suitably revised the manuscript to address the major technical and conceptual concerns noted with the initial submission.

As requested, they have added new experimental data defining the specificity of the interaction of tau with HDAC6 relative to at least one Class I HDAC member HDAC1. They have provided new data showing that only 1 of the 2 deacetylase domains of HDAC6 is involved in tau deacetylation with an elegant dissection using site-specific mutations (H216/611A) in the HDAC6 catalytic domain showing that only the C-terminal domain has tau deacetylase activity.

Additionally, the inclusion of new data from analysis of a MAPT human case with a P301L mutation showing that it is negative for AcK311 levels is a helpful clarification in terms of comparisons made to the PS19 mouse model that was shown to have AcK311 detectable but only upon knockout of HDAC6.

Although I had requested that the authors consider performing a long-term HDAC6 inhibitor treatment in PS19 mice to try to reconcile the observations of Selencia et al. using in Tg4510 mouse model the authors have cited recent findings pointing to limitation so the Tg4510 mouse as one possible explanation and described limitations with tubastatin A that makes the experiment challenging for long-term administration. They have now included new findings comparing the effects of HDAC6 knockout in a 5X FAD mutant background (5xFHK) and again see protection rather than an acceleration of tauopathy. **Taken together, and in light of experimental limitations due to COVID-19, I no longer think this experiment holds sufficient merit to hold up the publication of this experiment.**

Overall, the conclusion that the genetic loss of HDAC6 accelerates rather than alleviates tauopathy in mice is important. The authors have made a strong contribution to the field by their detailed and rigorously studies that ultimately inform current efforts focused on understanding the role of HDAC6 in protein quality control pathways and targeting HDAC6 therapeutically.

Minor points:

The substrate in the HDAC6 (BML-AK516) FLUOR DE LYS assay is in fact not specific for HDAC6 as stated in the response letter. As indicated by the manufacture (<https://www.enzolifesciences.com/BML-AK516/fluor-de-lys-hdac6-fluorometric-drug-discovery-kit/>) the substrate used in this kit is based upon the FLUOR DE LYS SIRT1, Deacetylase Substrate (Prod. No. BML-KI177), which is derived from residues 379-382 of p53 (Arg-His-Lys-Lys(Ac)), and is known to be deacetylated by a wide range of HDACs and sirtuins. This does not affect the conclusion of the authors.

We very much thank the reviewer for this comment; we have updated this point in the manuscript to reflect that the substrate is not HDAC6-specific.

Figure 4 is extremely dense and difficult to read given the size of figures and graphs.

We tried to simplify this figure and considered moving quantification results to the supplementary section, but all of this data is critical and relevant for interpretation of the figure. We feel it is best to show all of the quantification in detail, which was requested by the reviewers in our 1st revision. Therefore, since all panels provide useful primary information, we feel this should be kept as is.

REVIEWERS' COMMENTS:

Reviewer #1 (Remarks to the Author):

I appreciate the time the authors have taken to address my previous comments. I am now satisfied with the edits they have made to the manuscript and I am happy to recommend it for publication in Nature Communications.